# GOCor: Bringing Globally Optimized Correspondence Volumes into Your Neural Network

**Prune Truong**\*, **Martin Danelljan**\*, **Luc Van Gool, Radu Timofte**
{prune.truong, martin.danelljan, vangool, radu.timofte}@vision.ee.ethz.ch
Computer Vision Lab, ETH Zurich, Switzerland

## Abstract

The feature correlation layer serves as a key neural network module in numerous computer vision problems that involve dense correspondences between image pairs. It predicts a correspondence volume by evaluating dense scalar products between feature vectors extracted from pairs of locations in two images. However, this point-to-point feature comparison is insufficient when disambiguating multiple similar regions in an image, severely affecting the performance of the end task. We propose GOCor, a fully differentiable dense matching module, acting as a direct replacement to the feature correlation layer. The correspondence volume generated by our module is the result of an internal optimization procedure that explicitly accounts for similar regions in the scene. Moreover, our approach is capable of effectively learning spatial matching priors to resolve further matching ambiguities. We analyze our GOCor module in extensive ablative experiments. When integrated into state-of-the-art networks, our approach significantly outperforms the feature correlation layer for the tasks of geometric matching, optical flow, and dense semantic matching. The code and trained models will be made available at github.com/PruneTruong/GOCor.

## 1   Introduction

Finding pixel-wise correspondences between pairs of images is a fundamental problem in many computer vision domains, including optical flow [14, 19, 21, 28, 51, 52, 55], geometric matching [13, 37, 41, 42, 55], and disparity estimation [11, 33, 40, 61]. Most recent state-of-the-art approaches rely on feature correlation layers, evaluating dense pair-wise similarities between deep representations of two images. The resulting four-dimensional *correspondence volume* captures dense *matching confidences* between every pair of image locations. It serves as a powerful cue in the prediction of, for instance, optical flow. This encapsulation of dense correspondences has further achieved wide success within semantic matching [9, 12, 18, 23, 25, 26, 27, 43, 55], video object segmentation [10, 17, 39, 57], and few-shot segmentation [36, 58]. The feature correlation layer thus serves as a key building block when designing network architectures for a diverse range of important computer vision applications.

In the feature correlation layer, each confidence value in the correspondence volume is obtained as the scalar product between two feature vectors, extracted from specific locations in the two images, here called the *reference* and the *query* images. However, the sole reliance on point-to-point feature comparisons is often insufficient in order to disambiguate multiple similar regions in an image. As illustrated in Fig. 1, in the case of repetitive patterns, the feature correlation layer generates undistinctive and inaccurate matching confidences (Fig. 1d), severely affecting the performance of the end task. This remains the key limitation of feature correlation layers, since repetitive patterns, low-textured regions, and co-occurring similar objects are all pervasive in computer vision applications.

We design a new dense matching module, aiming to address the aforementioned issues by exploring information not exploited by the feature correlation layer. We observe that a confidence value in

---

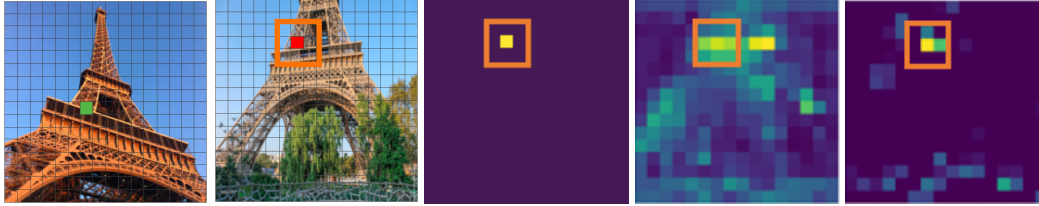

| (a) Reference image | (b) Query image | (c) Ideal Correlation | (d) Feat. Correlation | (e) GOCor (Ours) |

Figure 1: Visualization of the matching confidences (c)-(e) computed between the indicated location (green) in the reference image (a) and all locations of the query image (b). The feature correlation (d) generates undistinctive and inaccurate confidences due to similar regions and repetitive patterns. In contrast, our GOCor (e) predicts a distinct high-confidence value at the correct location.

the correspondence volume generated by the feature correlation layer only depends on the feature vectors extracted at one pair of locations in the reference and query. However, the reference also contains the appearance information of other image locations, that are likely to occur in the query image. This includes the appearance of similar regions in the scene, opening the opportunity to actively identify and account for such similarities when estimating each matching confidence value. Moreover, the feature correlation layer ignores prior knowledge and constraints that can be derived from the query, *e.g.* the uniqueness and spatial smoothness of correspondences. Our matching module encapsulates the aforementioned information and constraints into a learnable objective function. Our enhanced correspondence volume is obtained by minimizing this objective during the forward-pass of the network. This allows us to predict *globally optimised* correspondence volumes, effectively accounting for similar image regions and matching constraints, as visualized in Fig. 1e.

**Contributions:** We introduce GOCor, a differentiable neural network module that generates the correspondence volume between a pair of images, acting as a direct replacement to the feature correlation layer. Our main contributions are as follows. **(i)** Our module is formulated as an internal optimization procedure that minimizes a customizable matching-objective during inference, thereby providing a general framework for effectively integrating both explicit and learnable matching constraints. **(ii)** We propose a robust objective that integrates information about similar regions in the scene, allowing our GOCor module to better disambiguate such cases. **(iii)** We introduce a learnable objective for capturing constraints and prior information about the query frame. **(iv)** We apply effective unrolled optimization, paired with accurate initialization, ensuring efficient end-to-end training and inference. **(v)** We perform extensive experiments on the geometric matching and optical flow tasks by integrating our module into state-of-the-art network architectures. Our approach outperforms the feature correlation layer in terms of both accuracy and robustness. In particular, our GOCor module demonstrates better domain generalization properties.

## 2 Related work

**Enhancing the correlation volume:** Since the quality of the correspondence volume is of prime importance, several works focus on improving it using learned post-processing techniques [29, 31, 43, 60]. Notably, Rocco *et al.* [43] proposed a trainable neighborhood consensus network, NC-Net, applied after the correlation layer to filter out ambiguous matches. Instead, we propose a fundamentally different approach, operating directly on the underlying feature maps, *before* the correlation operation. Our work is also related to [24, 44], which generate filters dynamically conditioned on an input [24] or features updated with an attentional graph neural network, whose edges are defined within the same or the other image of a pair [44]. Xiao *et al.* [59] also recently introduced a learnable cost volume that adapts the features to an elliptical inner product space.

**Optimization-based meta-learning:** Our approach is related to optimization-based meta-learning [4, 5, 6, 30, 56, 62]. In fact, our GOCor module can be seen as an internal learner, which solves the regression problem defined by our objective. In particular, we adopt the steepest descent based optimization strategy shown effective in [5, 6]. From a meta-learning viewpoint, our approach however offers a few interesting additions to the standard setting. Unlike for instance, in few-shot classification [4, 30, 62] and tracking [5, 56], our learner constitutes an internal network module of a larger architecture. This implies that the output of the learner does not correspond to the final network output, and therefore does not receive direct supervision during (meta-)training. Lastly, our learner module actively utilizes the query sample through the introduced trainable objective function.

## 3 Method

### 3.1 Feature Correlation Layers

The feature correlation layer has become a key building block in the design of neural network architectures for a variety of computer vision tasks, which either rely on or benefit from the estimation of dense correspondences between two images. To this end, the feature correlation layer computes a dense set of scalar products between localized deep feature vectors extracted from the two images, in the form of a four-dimensional *correspondence volume*. We consider two deep feature maps $f^r = \phi(I^r)$ and $f^q = \phi(I^q)$ extracted by a deep CNN $\phi$ from the *reference* image $I^r$ and the *query* image $I^q$, respectively. The feature maps $f^r, f^q \in \mathbb{R}^{H \times W \times D}$ have a spatial size of $H \times W$ and dimensionality $D$. We let $f^r_{ij} \in \mathbb{R}^D$ denote the feature vector at a spatial location $(i, j)$. The feature correlation layer evaluates scalar products $(f^r_{ij})^{\mathrm{T}} f^q_{kl}$ between the reference and query image representations. There are two common variants of the correlation layer, both relying on the same local scalar product operation, but with some important differences. We define these operations next.

The **Global correlation layer** evaluates the pairwise similarities between all locations in the reference and query feature maps. This is defined as the operation,

$$\mathbf{C}_{\mathrm{G}}(f^r, f^q)_{ijkl} = (f^r_{ij})^{\mathrm{T}} f^q_{kl}, \quad (i, j), (k, l) \in \{1, \ldots, H\} \times \{1, \ldots, W\}. \tag{1}$$

The result is thus a 4D tensor $\mathbf{C}_{\mathrm{G}}(f^r, f^q) \in \mathbb{R}^{H \times W \times H \times W}$ capturing the similarities between all pairs of spatial locations. In the **Local correlation layer**, the scalar products involving $f^r_{ij}$ are instead only evaluated in a neighborhood of the location $(i, j)$ in the query feature map $f^q$,

$$\mathbf{C}_{\mathrm{L}}(f^r, f^q)_{ijkl} = (f^r_{ij})^{\mathrm{T}} f^q_{i+k,j+l}, \ (i, j) \in \{1, \ldots, H\} \times \{1, \ldots, W\}, \ (k, l) \in \{-R, \ldots, R\}^2. \tag{2}$$

$(k, l)$ represents the displacement relative to the reference frame location $(i, j)$, constrained to a value within the search radius $R$. While the limited search region $R$ makes the local correlation practical even for feature maps of a large spatial size $H \times W$, it does not capture similarities beyond $R$.

### 3.2 Motivation

The main purpose of feature correlation layers is to predict a dense set of matching confidences between the two images $I^r$ and $I^q$. This is performed in (1)-(2) by applying each reference frame feature vector $f^r_{ij}$ to a region in the query $f^q$. However, this operation ignores two important sources of valuable information when establishing dense correspondences.

**Reference frame information :** The matching confidences $\mathbf{C}(f^r, f^q)_{ij..} \in \mathbb{R}^{H \times W}$ (in 1-2) for the reference image location $(i, j)$ does not account for the appearance at other locations of the reference image. Instead, it only depends on the feature vector $f^r_{ij}$ at the location itself. This is particularly problematic when the reference frame contains multiple locations with similar appearance, such as repetitive patterns or homogeneous regions (see Fig. 1). These regions are also very likely to occur in the query feature map $f^q$, since it usually depicts the same scene at a later time instance or from a different viewpoint. This easily results in high correlation values at multiple incorrect locations, often severely affecting the accuracy and robustness of the final network prediction. Unfortunately, patterns of similar appearance are almost ubiquitous in natural scenes. Therefore, the estimation of matching confidences should ideally exploit the *known similarities in the reference image itself*.

**Query frame information :** The second source of information not exploited by the feature correlation layer is matching constraints and priors that can be derived from the query $f^q$. One such important constraint is that each reference image location $f^r_{ij}$ can have at most one matching location $f^q_{kl}$ in the query image. Moreover, dense matches across the image pair generally follow spatial smoothness properties, due to the spatio-temporal continuity of the underlying 3D scene. This can serve as a powerful prior when predicting the correspondence volume between the image pair.

Next, we set out to develop a dense matching module capable of effectively utilizing the aforementioned information when predicting the correspondence volume relating $I^r$ and $I^q$.

### 3.3 General Formulation

In this section, we formulate GOCor, an end-to-end differentiable neural network module capable of generating more accurate correspondence volumes than feature correlation layers. We start by

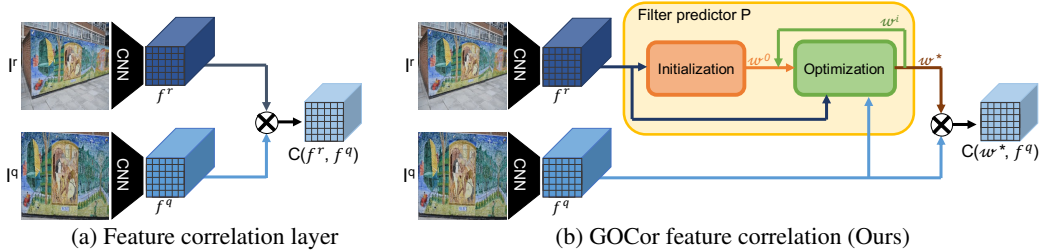

(a) Feature correlation layer          (b) GOCor feature correlation (Ours)

Figure 2: Schematic overview of the the feature correlation layer (a) and our GOCor module (b).

replacing the reference feature map $f^r$ in (1)-(2) with a general tensor $w^*$ of the same size, which we refer to as the *filter map*. Instead of correlating the reference features $f^r$ with the query $f^q$, we aim to first predict the filter map $w^*$, enriched with the global information about the reference $f^r$ and query $f^q$ described in the previous section. The filter map $w^*$ is then applied to the query features $f^q$ to obtain the final correspondence volume as $\mathbf{C}(w^*, f^q)$. We use $\mathbf{C}$ to denote either global (1) or local (2) correlation. We thus embrace the correlation operation (1)-(2) itself, and aim to enhance its output by enriching its input.

The remaining part of our method description is dedicated to the key question raised by the above generalization, namely how to achieve a suitable filter map $w^*$. In general, we can consider it to be the result of a differentiable function $w^* = P_\theta(f^r, f^q)$, which takes the reference and query features as input and has a set of trainable parameters $\theta$. For example, simply letting $P_\theta(f^r, f^q) = f^r$ retrieves the original feature correlation layer $\mathbf{C}(f^r, f^q)$. However, designing a neural network module $w^* = P_\theta(f^r, f^q)$ that *effectively* takes advantage of the information and constraints discussed in Sec. 3.2 is challenging. Moreover, we require our module to robustly generalize to new domains, having image content and motion patterns not seen during training.

We tackle these challenges by formulating an objective function $L$, that explicitly encodes the constraints discussed in Sec. 3.2. The network module $P_\theta(f^r, f^q)$ is then constructed to output the filter map $w^*$ that minimizes this objective,

$$w^* = P_\theta(f^r, f^q) = \arg\min_w L(w; f^r, f^q, \theta). \tag{3}$$

This formulation allows us to construct the filter predictor module $P_\theta$ by designing an objective $L$ along with a suitable optimization algorithm. It gives us a powerful framework to explicitly integrate the constraints discussed in Sec. 3.2, while also benefiting from significant interpretability. In the next sections, we formulate our objective function $L$. We first integrate information about the reference features $f^r$ into the objective (3) in Sec. 3.4. In Sec. 3.5, we then extend the objective $L$ with information about the query $f^q$. Lastly, we discuss the optimization procedure applied to our objective in Sec. 3.6. An overview of our general matching module is illustrated in Figure 2.

## 3.4 Reference Frame Objective

Here, we introduce a flexible objective that exploits global information about the reference features $f^r$, as discussed in Sec. 3.2. For convenience, we follow the convention for global correlation (1) by letting subscripts denote absolute spatial locations. When establishing matching confidences for a reference frame location $(i, j)$, the feature correlation layer $\mathbf{C}(f^r, f^q)$ only utilizes the encoded appearance $f^r_{ij}$ at the location $(i, j)$. However, the reference feature map

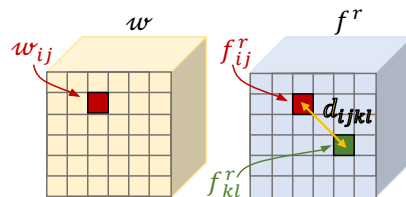

Figure 3: Visualization of the filter map $w$ and reference feature map $f^r$.

$f^r$ also contains the encoding $f^r_{kl}$ of other image regions $(k, l)$, which are likely to also occur in the query $f^q$. To exploit this information, we therefore first replace the reference feature map $f^r$ with our filter map $w$. The aim is then to find $w$ which enforces high confidences $\mathbf{C}(w, f^r)_{ijij} = w^T_{ij} f^r_{ij} \approx 1$ at the corresponding reference location $(i, j)$, while ensuring low matching confidences $\mathbf{C}(w, f^r)_{ijkl} = w^T_{ij} f^r_{kl} \approx 0$ for other locations $(k, l) \neq (i, j)$ in the reference map $f^r$. These constraints aim at designing $w_{ij}$, that explicitly suppresses the corresponding matching confidences in regions $f^r_{kl}$ that have similar appearance as $f^r_{ij}$, since these regions may also occur in $f^q$.

As a first attempt, the aforementioned reference-frame constraints could be realized by minimizing the quadratic objective $\|\mathbf{C}(w, f^r) - \delta\|^2$. Here, $\delta$ represents the desired correlation response, which

in case of global correlation (1) is $\delta_{ijkl} = 1$ whenever $(i,j) = (k,l)$ and $\delta_{ijkl} = 0$ otherwise. The quadratic objective is attractive since it can be tackled with particularly effective optimization methods. On the other hand, the simple quadratic objective is known for its sensitivity to outliers. In our setting, the objective should in fact be largely indifferent to cases when a non-matching pair generates a strong negative correlation output $w_{ij}^{\mathsf{T}} f_{kl}^{r} \ll 0$. This stems from the fact that any zero *or* negative confidence is enough to indicate a non-match. However, such strong negative predictions receive a disproportionately large impact in the quadratic objective, instead compromising the quality of the correspondence volume in challenging regions with similar appearance. This issue is further amplified by the severe imbalance between examples of *matches* and *non-matches* in the objective.

To address these issues, we formulate a robust non-linear least squares objective. For a non-matching location pair ($\delta_{ijkl} = 0$), a positive correlation output $\mathbf{C}(w, f^r)_{ijkl} > 0$ corresponds to a similar appearance that should be suppressed, while negative correlation output $\mathbf{C}(w, f^r)_{ijkl} < 0$ is of little importance. We account for this asymmetry by introducing separate penalization weights $v_{ijkl}^{+}$ and $v_{ijkl}^{-}$ for positive and negative correlation outputs, respectively. The confidence values are thus mapped by the scalar function $\sigma$ defined as,

$$\sigma(c; v^+, v^-) = \begin{cases} v^+ c, & c \geq 0 \\ v^- c, & c < 0 \end{cases}, \tag{4a}$$

$$\sigma_\eta(c; v^+, v^-) = \frac{v^+ - v^-}{2}\left(\sqrt{c^2 + \eta^2} - \eta\right) + \frac{v^+ + v^-}{2}c. \tag{4b}$$

We have also defined a smooth approximation $\sigma_\eta$, which for $\eta > 0$ avoids the discontinuity in the derivative of $\sigma$ at $\mathbf{C}(w, f^r) = 0$. The original function $\sigma = \sigma_0$ is retrieved by setting $\eta = 0$.

By applying the function (4), the confidence values $\mathbf{C}(w, f^r)$ can be re-weighted using appropriate values for the weights $v^+$ and $v^-$. To address the question of how to set $v^+$ and $v^-$ in practice, recall that our objective defines a neural network module through the optimization (3). This opens an interesting opportunity of learning $v^+$ and $v^-$ as parameters of the neural network. These can thus be trained along with all other parameters of the network for the end task. Specifically, we parametrize the weights as functions $v_{ijkl}^{+} = v_\theta^{+}(d_{ijkl})$ and $v_{ijkl}^{-} = v_\theta^{-}(d_{ijkl})$ of the distance $d_{ijkl} = \sqrt{(i-k)^2 + (j-l)^2}$ between $w_{ij}$ and the example $f_{kl}^{r}$. This strategy allows the network to learn the transition between the correct match $d_{ijij} = 0$ and the distant $d_{ijkl} \gg 0$ examples of non-matching features $f_{kl}^{r}$. Our robust and learnable objective function for integrating reference frame information is thus formulated as,

$$L_{\mathrm{r}}(w; f^r, \theta) = \left\| \sigma_\eta\big(\mathbf{C}(w, f^r); v^+, v^-\big) - y \right\|^2. \tag{5}$$

Here we have additionally replaced the ideal correlation $\delta$ with a learnable target confidence $y_{ijkl} = y_\theta(d_{ijkl})$, to add further flexibility. We parametrize $v_\theta^{+}$, $v_\theta^{-}$, and $y_\theta$ using the strategy introduced in [5], as piece-wise linear functions of the distance $d_{ijkl}$, further detailed in the appendix, Sec. C.

## 3.5 Query Frame Objective

In the previous section, we formulated an approach that integrates the reference feature map $f^r$ into the objective (3). However, as discussed in Sec. 3.2, there is also rich information to gain from the query frame. Firstly, correspondences between a pair of images must adhere to certain constraints, mainly that each point in the reference image can have at most a single match in the query image. Secondly, neighboring matches follow spatial smoothness priors, largely induced by the spatio-temporal continuity of the underlying 3D-scene. We encapsulate such constraints by defining a regularizing objective on the query frame,

$$L_{\mathrm{q}}(w; f^q, \theta) = \left\| R_\theta * \mathbf{C}(w, f^q) \right\|^2. \tag{6}$$

Here, $*$ denotes the convolution operator and $R_\theta \in \mathbb{R}^{K^4 \times Q}$ is a learnable 4D-kernel of spatial size $K$ and $Q$ number of output channels. A 4D-convolution operator allows us to fully utilize the structure of the 4D correspondence volume. Furthermore, its use is motivated by the translation invariance property induced by the 2D translation invariance of the two input feature maps. $R_\theta$ is learnt, along with all other network parameters, by the SGD-based minimization of the final network training loss.

The use of smoothness priors has a long and successful tradition in classic variational formulations for optical flow, developed during the pre-deep learning era [2, 7, 16, 35]. We therefore take inspiration

from these ideas. However, our approach offers several interesting conceptual differences. First, our regularization operates directly on the matching confidences generated by the correlation operation, rather than the flow vectors. The correspondence volume provides a much richer description by encapsulating uncertainties in the correspondence assignment. Second, our objective is a function of the underlying filter map $w$, which is the input to the correlation layer. Third, our objective is implicitly minimized *inside* a deep neural network. Finally, this further allows our regularizer $R_\theta$ to be learned in a fully end-to-end and data-driven manner. In contrast, classical methods rely on hand-crafted regularizers and priors. By integrating information from a local 4D-neighborhood, the operator $R_\theta$ in (6) can enforce spatial smoothness by, for instance, learning differential operators. Moreover, our formulation lets the network learn the weighting of the query term (6) in relation to the reference frame objective (5), eliminating the need for such hyper-parameter tuning.

### 3.6 Filter map prediction module $P$

Our objective, employed in (3), is obtained by combining the reference (5) and query (6) terms as,

$$L(w; f^r, f^q, \theta) = L_\mathrm{r}(w; f^r, \theta) + L_\mathrm{q}(w; f^q, \theta) + \|\lambda_\theta w\|^2 \,. \tag{7}$$

The last term corresponds to a regularizing prior on $w$, weighted by the learnable scalar $\lambda_\theta \in \mathbb{R}$. Note that while the reference frame objective $L_\mathrm{r}$ in (5) can be decomposed into independent terms for each location $w_{ij}$, the query term $L_\mathrm{q}$ (6) introduces dependencies between all elements in $w$. Efficiently optimizing such a high-dimensional problem during the forward pass of the network in order to implement (3) may seem an impossibility. Next, we demonstrate that this can, in fact, be achieved by a combination of accurate initialization and a simple but powerful iterative procedure. Any neural network architecture employing feature correlation layers can thereby benefit from our module.

**Optimizer:** While finding the global optima of (7) within a small tolerance is costly, this is not necessary in our case. Instead, we can effectively utilize the information encoded in the objective (7) by optimizing it to *a sufficient degree*. We therefore derive the filter map $w^* = P_\theta(f^r, f^q)$ by applying an iterative optimization strategy. Specifically, we use the Steepest Descent algorithm, which was found effective in [5]. Given the current iterate $w^n$, the steepest descent method [38, 49] finds the step-length $\alpha^n$ that minimizes the objective in the gradient direction. This is obtained through a simple closed-form expression by first performing a Gauss-Newton approximation of (5). The filter map is then updated by taking a gradient step with optimal length $\alpha^n$,

$$w^{n+1} = w^n - \alpha^n \nabla L\left(w^n; f^r, f^q, \theta\right) \,, \quad \alpha^n = \arg\min_\alpha L_\mathrm{GN}^n\left(w^n - \alpha \nabla L(w^n; f^r, f^q, \theta)\right) \,. \tag{8}$$

Here, $L_\mathrm{GN}^n$ is the Gauss-Newton approximation of (7) at $w^n$. Both the gradient $\nabla L$ and the step length $\alpha^n$ are implemented using their closed form expressions with standard neural network modules, as detailed in the appendix Sec. A. Importantly, the operation (8) is fully differentiable w.r.t. $f^r$, $f^q$, and $\theta$, allowing end-to-end training of all underlying network parameters.

**Initializer:** To reduce the number of optimization iterations needed in the filter predictor network $P$, we generate an initial filter map $w^0$ using an efficient and learnable module. We parametrize $w_{ij}^0 = a_{ij} f_{ij}^r + b_{ij} \bar{f}^r$, where $\bar{f}^r \in \mathbb{R}^d$ is the spatial average reference vector, encoding contextual information. Intuitively, we wish $w^0$ to have a high activation $(w_{ij}^0)^\mathrm{T} f_{ij}^r = 1$ at the matching position and $(w_{ij}^0)^\mathrm{T} \bar{f}^r = 0$. The scalar coefficients $a_{ij}$ and $b_{ij}$ are then easily found by solving these equations. Details are given in the appendix Sec. B.

## 4 Experiments

We perform comprehensive experiments for two tasks: geometric correspondences and optical flow. We additionally show that our method can be successfully applied to the task of semantic matching. Both global and local correlation-based versions of our GOCor module are analyzed by integrating them into two recent state-of-the-art networks. Further results, analysis, and visualizations along with more details regarding architectures and datasets are provided in the appendix.

### 4.1 Geometric matching

We first evaluate our GOCor module for dense geometric matching by integrating it into the recent GLU-Net [55]. GLU-Net is a 4-level pyramidal network, operating at two image resolutions to

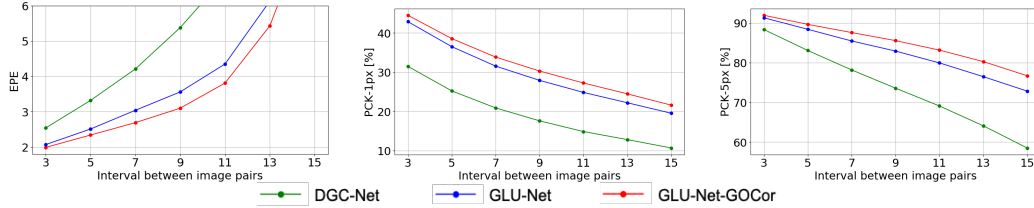

Figure 4: Results on geometric matching dataset ETH3D [47]. AEPE (left), PCK-1 (center), and PCK-5 (right) are plotted w.r.t. the inter-frame interval length.

estimate dense flow fields. It relies on a global correlation at the coarsest level to capture long-range displacements and uses local correlations in the subsequent levels.

**Experimental setup:** We create GLU-Net-GOCor by replacing global and local feature correlation layers with our global and local GOCor modules, respectively. The global GOCor module employs the full objective (7), while the local variant uses only the reference term (5). We use three steepest descent iterations during training and increase the number during inference. We follow the same self-supervised training procedure and data as in [55], applying synthetic homography transformations to images compiled from different sources to ensure diversity. We refer to this as the *Static* dataset, since it simulates a static scene. For better compatibility with real 3D scenes and moving objects, we further introduce a *Dynamic* training dataset, by augmenting the *Static* data with random independently moving objects from the COCO [34] dataset. In all experiments, we compare the results of GLU-Net and GLU-Net-GOCor trained with the *same* data, and according to the *same* procedure.

**Evaluation datasets and metrics:** We first employ the 59 sequences of the **HPatches** dataset [3], consisting of planar scenes from different viewpoints. We additionally utilize the multi-view **ETH3D** dataset [47], depicting indoor and outdoor scenes captured from a moving hand-held camera. We follow the protocol of [55], sampling image pairs at different intervals to analyze varying magnitude of geometric transformations. Finally, because of the difficulty to obtain dense annotations on real imagery with extreme viewpoint and varying imaging condition, we also evaluate our model on sparse correspondences available on the **MegaDepth** [32] dataset, according to the protocol introduced in [48]. We use the *Static* training data for the comparison on the HPatches dataset and the *Dynamic* training data for the ETH3D and MegaDepth datasets. In line with previous works [37, 55], we employ the Average End-Point Error (AEPE) and Percentage of Correct Keypoints at a given pixel threshold $T$ (PCK-$T$) as the evaluation metrics.

**Results:** In Table 1, we present results on HPatches. We also report the results of the recent state-of-the-art DGC-Net [37] for reference. Our GLU-Net-GOCor outperforms original GLU-Net by a large margin, achieving both higher accuracy

Table 1: HPatches homography dataset [3].

|  | AEPE ↓ | PCK-1 (%) ↑ | PCK-5 (%) ↑ |
|---|---|---|---|
| DGC-Net [37] | 33.26 | 12.00 | 58.06 |
| GLU-Net | 25.05 | 39.55 | 78.54 |
| GLU-Net-GOCor (Ours) | **20.16** | **41.55** | **81.43** |

in terms of PCK, and better robustness to large errors as indicated by AEPE. In Figure 4, we plot AEPE, PCK-1 and PCK-5 obtained on the ETH3D images. For all intervals, our approach is consistently better than baseline GLU-Net. We note that the improvement is particularly prominent at larger intra-frame intervals, strongly indicating that our GOCor module better copes with large appearance variations due to large viewpoint changes, compared to the feature correlation layer.

Table 2: Results on sparse correspondences of the MegaDepth dataset [32].

|  | PCK-1 (%) ↑ | PCK-3 (%) ↑ | PCK-5 (%) ↑ |
|---|---|---|---|
| GLU-Net | 21.58 | 52.18 | 61.78 |
| GLU-Net-GOCor (Ours) | **37.28** | **61.18** | **68.08** |

This is also confirmed by the results on MegaDepth in Table 2. Images depict extreme view-point changes with as little as 10% of co-visible regions. In this case as well, GOCor brings significant improvement, particularly in pixel-accuracy (PCK-1).

### 4.2 Optical flow

Next, we evaluate our GOCor module for the task of optical flow estimation, by integrating it into the state-of-the-art PWC-Net [51, 52] and GLU-Net [55] architectures. PWC-Net [51] is based on a 5-level pyramidal network, estimating the dense flow field at each level using a local correlation layer.

**Experimental setup:** We replace all local correlation layers with our local GOCor module to obtain PWC-Net-GOCor. We finetune PWC-Net-GOCor on *3D-Things* [21], using the publicly available PWC-Net weights trained on *Flying-Chairs* [14] and *3D-Things* [21] as initialization. For

Table 3: Results for the optical flow task on the training splits of KITTI [15] and Sintel [8]. A result in parenthesis indicates that the dataset was used for training.

| | KITTI-2012 | | KITTI-2015 | | Sintel Clean | | | Sintel Final | | |
|---|---|---|---|---|---|---|---|---|---|---|
| | AEPE $\downarrow$ | F1 (%) $\downarrow$ | AEPE $\downarrow$ | F1 (%) $\downarrow$ | AEPE $\downarrow$ | PCK-1 (%) $\uparrow$ | PCK-5 (%) $\uparrow$ | AEPE $\downarrow$ | PCK-1 (%) $\uparrow$ | PCK-5 (%) $\uparrow$ |
| GLU-Net | 3.14 | 19.76 | 7.49 | 33.83 | 4.25 | 62.08 | 88.40 | 5.50 | 57.85 | 85.10 |
| GLU-Net-GOCor | **2.68** | **15.43** | **6.68** | **27.57** | **3.80** | **67.12** | **90.41** | **4.90** | **63.38** | **87.69** |
| PWC-Net (from paper) | 4.14 | 21.38 | 10.35 | 33.67 | 2.55 | - | - | 3.93 | - | - |
| PWC-Net (*ft 3D-Things*) | 4.34 | 20.90 | 10.81 | 32.75 | 2.43 | 81.28 | 93.74 | 3.77 | 76.53 | 90.87 |
| PWC-Net-GOCor (*ft 3D-Things*) | **4.12** | **19.31** | **10.33** | **30.53** | **2.38** | **82.17** | **94.13** | **3.70** | **77.34** | **91.20** |
| PWC-Net (*ft Sintel*) | 2.94 | 12.70 | 8.15 | 24.35 | (1.70) | - | - | (2.21) | - | - |
| PWC-Net-GOCor (*ft Sintel*) | **2.60** | **9.67** | **7.64** | **20.93** | (1.74) | (87.93) | (95.54) | (2.28) | (84.15) | (93.71) |

fair comparison, we also finetune the standard PWC-Net on *3D-Things* with the same schedule. Finally, we also finetune PWC-Net-GOCor on the *Sintel* [8] training dataset according to the schedule introduced in [21, 51]. As described in Sec. 4.1, we train both GLU-Net and GLU-Net-GOCor on the *Dynamic* training set. For the global and local GOCor modules, we use the same settings as in 4.1.

**Datasets and evaluation metrics:** For evaluation, we use the established **KITTI** dataset [15], composed of real road sequences captured by a car-mounted stereo camera rig. We also utilize the **Sintel** dataset [8], which consists of 3D animated sequences. We use the standard evaluation metrics, namely the AEPE and F1 for KITTI. The latter represents the percentage of optical flow outliers. For Sintel, we employ AEPE together with PCK, *i.e.* percentage of inliers. In line with [19, 20, 51, 52, 55], we show results on the training splits of these datasets.

**Results:** Results are reported in Tab. 3. First, compared to the GLU-Net baseline, our GOCor module brings significant improvements in both AEPE and F1/PCK on all optical flow datasets. Next we compare the PWC-Net based methods trained on *3D-Things* (middle section) and report the official result [51, 52] along with our fine-tuned versions. While our PWC-Net-GOCor obtains a similar AEPE, it achieves substantially better accuracy, with a 3% improvement in F1 metric on KITTI-2015. After finetuning on Sintel images, both PWC-Net and PWC-Net-GOCor achieve similar results on the Sintel training data (in parenthesis). However, the PWC-Net-GOCor version provides superior results on the two KITTI datasets. This clearly demonstrates the superior domain generalization capabilities of our GOCor module. Note that both methods in the bottom section of Tab. 3 are only trained on animated datasets, while KITTI consists of natural road-scenes. Thanks to the effective objective-based adaption performed in our matching module during inference, PWC-Net-GOCor excels even with a sub-optimal feature embedding trained for animated images, and when exposed to previously unseen motion patterns. This is a particularly important property in the context of optical flow and geometric matching, where collection of labelled realistic training data is prohibitively expensive, forcing methods to resort to synthetic and animated datasets.

## 4.3 Generalization to semantic matching

We additionally compare the performance of GOCor to the feature correlation layer on the task of semantic matching. In Table 4, we evaluate our GLU-Net-GOCor, without any re-training, for dense semantic matching on the TSS dataset [54]. In the semantic correspondence task, images depict different

Table 4: PCK [%] on TSS.

| | FGD3Car | JODS | PASCAL | All |
|---|---|---|---|---|
| Semantic-GLU-Net [55] | 94.4 | 75.5 | 78.3 | 82.8 |
| GLU-Net | 93.2 | 73.3 | 71.1 | 79.2 |
| GLU-Net-GOCor | **95.0** | **78.9** | **81.3** | **85.1** |

instances of the same object category (e.g. *horse*). As a result, the value of additional reference frame information (Sec. 3.2 and 3.4) is not as pronounced in semantic matching compared to geometric matching or optical flow. Indeed, our reference frame objective uses its full potential when both the reference and the query images depict similar regions from the *same scene*. Nevertheless, our GLU-Net-GOCor sets a new state-of-the-art on this dataset, even outperforming Semantic-GLU-Net [55].

## 4.4 Run-time

In Table 5, we compare the run time of our GOCor-based networks to their original versions on the KITTI-2012 dataset. The timings are obtained on the same desktop with an NVIDIA Titan X GPU. While our GOCor module leads to increased computation, the run-time remains within reasonable margins thanks to our dedicated optimization module, described in Sec. 3.6. We can further control the trade of between computation and performance by varying the number of steepest descent iterations in our GOCor module. In

Table 5: Run time [ms] averaged over the 194 image pairs of KITTI-2012.

| | Run-time [ms] |
|---|---|
| PWC-Net | 118.05 |
| PWC-Net-GOCor | 203.02 |
| GLU-Net | 154.97 |
| GLU-Net-GOCor | 261.90 |

Table 6: Ablation study of key aspects of our approach on three different datasets.

| | | HPatches | | KITTI-2012 | | KITTI-2015 | |
|---|---|---|---|---|---|---|---|
| | | AEPE $\downarrow$ | PCK-5 (%) $\uparrow$ | AEPE $\downarrow$ | F1 (%) $\downarrow$ | AEPE $\downarrow$ | F1 (%) $\downarrow$ |
| **(I)** | BaseNet | 30.94 | 69.22 | 4.03 | 30.49 | 8.93 | 48.66 |
| **(II)** | BaseNet + NC-Net [43] | 39.15 | 63.52 | 4.41 | 34.78 | 9.86 | 52.78 |
| **(III)** | BaseNet + Global-GOCor Linear Regression | 27.02 | 68.12 | 4.31 | 35.30 | 8.93 | 52.64 |
| **(IV)** | BaseNet + Global-GOCor $L_r$ | 26.27 | 71.29 | 3.91 | 29.77 | 8.50 | 46.24 |
| **(V)** | BaseNet + Global-GOCor $L_r + L_q$ | 25.30 | 71.21 | 3.74 | 26.82 | 7.87 | 43.08 |
| **(VI)** | BaseNet + Global-GOCor $L_r + L_q$ + Local-GOCor | **23.57** | **78.30** | **3.45** | **25.42** | **7.10** | **39.57** |

Appendix Sec. E.1 we provide such a detailed analysis, and propose faster operating points with only minor degradation in performance.

## 4.5 Ablation study

Finally, we analyze key components of our approach. We first design a powerful baseline architecture estimating dense flow fields, called BaseNet. It consists of a three-level pyramidal CNN-network, inspired by [55], employing a global correlation layer followed by two local layers. All methods are trained with the *Dynamic* data, described in Sec. 4.1. Results on HPatches, KITTI-2012 and KITTI-2015 are reported in Tab. 6. We first analyse the effect of replacing the feature correlation layer with GOCor at the global correlation level. The version denoted **(IV)** employs our global GOCor using solely the reference-based objective $L_r$ (Sec. 3.4). It leads to significantly better results on all datasets compared to standard BaseNet **(I)**. Instead of our robust reference loss $L_r$, the version **(III)** employs a standard linear regression objective $\|\mathbf{C}(w, f^r) - \delta\|^2$, leading to substantially worse results. We also compare with adding the post-processing strategy proposed in [43] **(II)**, employing 4D-convolutions and enforcing cyclic consistency. This generally leads to a degradation in performance, likely caused by the inability to cope with the domain gap between training and test data. From **(IV)** to **(V)** we integrate our query frame objective $L_q$ (Sec. 3.5), which results in major gains, particularly on the more challenging KITTI datasets. Finally, we replace the local correlation layers with our local GOCor module in **(VI)**. This leads to large improvements on all datasets and metrics.

In Figure 5, we visualize the relevance of our reference loss (Sec. 3.4) qualitatively by plotting the correspondence volume outputted by our global GOCor module, when correlating a particular point (i,j) of the reference image with all locations of either *the reference itself or the query image*. The predicted correspondence volume gets increasingly distinctive after each iteration in the GOCor layer. Specifically, it is clearly visible that final matching confidences with the query image benefits from optimizing the correlation scores with the reference image itself, using Eq. (5).

## 5 Conclusion

We propose a neural network module for predicting globally optimized matching confidences between two deep feature maps. It acts as a direct alternative to feature correlation layers. We integrate unexploited information about the reference and query frames by formulating an objective function, which is minimized during inference through an iterative optimization strategy. Our approach thereby explicitly accounts for, *e.g.*, similar image regions. Our resulting GOCor module is thoroughly analysed and evaluated on the tasks of geometric correspondences and optical flow, with an extension to dense semantic matching. When integrated into state-of-the-art networks, it significantly outperforms the feature correlation layer.

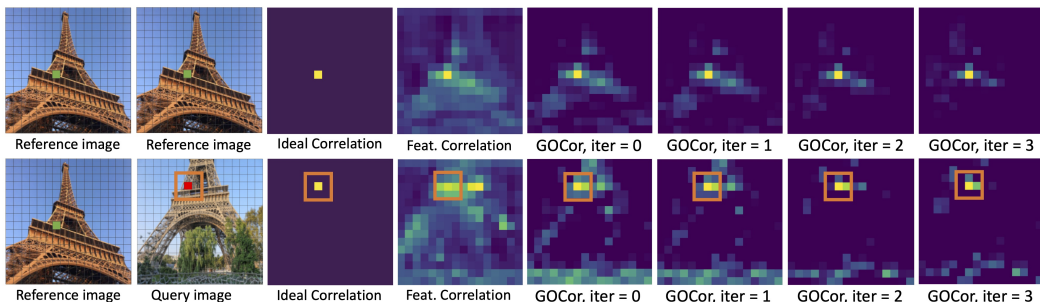

Figure 5: Visualization of the matching confidences computed between the indicated location (green) in the reference image and all locations of either the reference image itself or the query image.

## Broader Impact

Our feature correspondence matching module can be beneficial in a wide range of applications relying on explicit or implicit matching between images, such as visual localization [46, 53], 3D-reconstruction [1], structure-from-motion [45], action recognition [50] and autonomous driving [22]. On the other hand, any image matching algorithm runs the risk of being used for malevolent tasks, such as malicious image manipulation or image surveillance system. However, our module is only one building block to be integrated in a larger pipeline. On its own, it therefore has little chances of being wrongfully used.

## Acknowledgments and Disclosure of Funding

This work was partly supported by the ETH Zürich Fund (OK), a Huawei Technologies Oy (Finland) project, an Amazon AWS grant, and an Nvidia hardware grant.

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
