[Supplementary Material]

# Appendix

## GOCor: Bringing Globally Optimized Correspondence Volumes into Your Neural Network

**Prune Truong**[*], **Martin Danelljan**[*], **Luc Van Gool, Radu Timofte**
{prune.truong, martin.danelljan, vangool, radu.timofte}@vision.ee.ethz.ch
Computer Vision Lab, ETH Zurich, Switzerland

In this appendix, we first provide the details of the derivation of the filter map $w^*$ within the filter map predictor module $P$ in Section A. We then give the expression for the initial estimate $w^0$ in Section B. In Section C, we provide further insights on the architecture of our GOCor module as well as the implementation details. In Section D, we give more details about the evaluation datasets, metrics and networks utilized. We then provide additional quantitative and qualitative results of our approach GOCor compared to the feature correlation layer in Section E. Finally, we further analyse our approach in an extended ablation study in Section F.

## A  Derivation of filter map prediction module $P$

Here, we derive the iterative updates employed in our module $P_\theta$, which aims to solve $w^* = P_\theta(f^r, f^q) = \arg\min_w L(w; f^r, f^q, \theta)$ (Eq. 3 of the main paper). Our final objective (Eqs. 5-7 of the main paper) is given by,

$$L(w; f^r, f^q, \theta) = L_{\mathrm{r}}(w; f^r, \theta) + L_{\mathrm{q}}(w; f^q, \theta) + \|\lambda_\theta w\|^2 \tag{1a}$$

$$L_{\mathrm{r}}(w; f^r, \theta) = \left\|\sigma_\eta\big(\mathbf{C}(w, f^r); v^+, v^-\big) - y\right\|^2 \tag{1b}$$

$$L_{\mathrm{q}}(w; f^q, \theta) = \|R_\theta * \mathbf{C}(w, f^q)\|^2 . \tag{1c}$$

As discussed in Sec. 3.6 of the main paper, we do not need to attain a global optimum. The goal is to significantly minimize the loss $L$, using only a few iterations for efficiency. To this end we employ the Steepest Descent methodology [14, 20]. In the steepest descent algorithm, we update the parameters by taking steps $w^{n+1} = w^n - \alpha^n \nabla L(w^n)$ in the gradient direction $\nabla L(w^n)$ with step length $\alpha^n$. The aim is to find the step length $\alpha^n$ that leads to a maximal decrease in the objective. This is performed by first approximating the loss with a quadratic function at the current estimate $w^n$,

$$L(w) \approx L_{\mathrm{GN}}^n(w) = \frac{1}{2}(w - w^n)^{\mathrm{T}} Q^n (w - w^n) + (w - w^n)^{\mathrm{T}} \nabla L(w^n) + L(w^n) \tag{2}$$

Here, we see $w^n$ as a vector. We set the Hermitian positive definite matrix $Q^n$ according to the Gauss-Newton method [14] $Q^n = (J^n)^{\mathrm{T}} J^n$, where $J^n$ is the Jacobian of the residual at $w^n$. To avoid clutter, the dependence on $f^r, f^q, \theta$ is made implicit. In the rest of the section, unless otherwise stated, matrices multiplications are element-wise.

The steepest descent method [14, 20] finds the step-length $\alpha^n$ that minimizes the loss (2) in the gradient direction. Due to the convexity of (2), this is obtained by solving $\frac{\mathrm{d}}{\mathrm{d}\alpha} L_{\mathrm{GN}}(w^n - \alpha \nabla L(w^n)) = 0$, which leads to the expression,

$$\alpha^n = \frac{\nabla L(w^n)^{\mathrm{T}} \nabla L(w^n)}{\nabla L(w^n)^{\mathrm{T}} Q^n \nabla L(w^n)} = \frac{\|\nabla L(w^n)\|^2}{\|J^n \nabla L(w^n)\|^2} \tag{3}$$

In the next subsections, we derive the expression for $\nabla L$ and subsequently for step-length $\alpha$.

---

[*]Both authors contributed equally

## A.1 Closed form expression of $\nabla L$

Here, we derive a closed-form expression for the gradient of the loss (1a). The gradient $\nabla L(w)$ of the loss (1a) with respect to the filters $w$ is then computed as,

$$\nabla L(w) = \nabla L_{\mathrm{r}}(w) + \nabla L_{\mathrm{q}}(w) + 2\lambda_\theta^2 w\,. \tag{4}$$

**Expression of $\nabla L_{\mathbf{r}}(w)$:** $L_{\mathrm{r}}$ is defined according to (1b) and equation (5) of the main paper, such as $L_r = \|r_r(w, f^r)\|^2$. Here, $r_r$ designates the residual function, which is formulated as (also Eq. 4 of the main paper),

$$r_r(w, f^r) = \sigma_\eta\big(\mathbf{C}(w, f^r);\, v^+, v^-\big) - y \tag{5}$$

$$\sigma_\eta\big(\mathbf{C}(w, f^r);\, v^+, v^-\big) = \frac{v^+ - v^-}{2}\left(\sqrt{\mathbf{C}(w, f^r)^2 + \eta^2} - \eta\right) + \frac{v^+ + v^-}{2}\mathbf{C}(w, f^r)\,. \tag{6}$$

The gradient of $\nabla L_{\mathrm{r}}(w)$ of the loss (1b) w.r.t $w$ is given by:

$$\nabla L_{\mathrm{r}}(w) = 2\left[\frac{\partial r_r(w, f^r)}{\partial w}\right]^{\mathrm{T}} r_r(w, f^r) \tag{7}$$

where $J_r = \frac{\partial r_r(w, f^r)}{\partial w}$ corresponds to the Jacobian of the residual function (5) with respect to filters w. Using the chain rule we obtain,

$$J_r = \frac{\partial r_r(w, f^r)}{\partial w} = \frac{\partial \sigma_\eta}{\partial \mathbf{C}(w, f^r)}\frac{\partial \mathbf{C}(w, f^r)}{\partial w}\,, \tag{8}$$

Using (6), the derivative of the error function is obtained as

$$\frac{\partial \sigma_\eta}{\partial \mathbf{C}(w, f^r)} = \left[\frac{v^+ - v^-}{2}\left(\frac{\mathbf{C}(w, f^r)}{\sqrt{\mathbf{C}(w, f^r)^2 + \eta^2}}\right) + \frac{v^+ + v^-}{2}\right]\,. \tag{9}$$

Integrating (8) into (7) leads to the final formulation of $\nabla L_{\mathrm{r}}(w)$ as

$$\nabla L_{\mathrm{r}}(w) = 2\left[\frac{\partial \mathbf{C}(w, f^r)}{\partial w}\right]^{\mathrm{T}}\left[\left(\frac{v^+ - v^-}{2}\left(\frac{\mathbf{C}(w, f^r) - y}{\sqrt{(\mathbf{C}(w, f^r) - y)^2 + \eta^2}}\right) + \frac{v^+ + v^-}{2}\right) r_r(w, f^r)\right]\,. \tag{10}$$

The multiplication with the transposed Jacobian $\left[\frac{\partial \mathbf{C}(w, f^r)}{\partial w}\right]^{\mathrm{T}}$ corresponds to back-propagation through the correlation layer $\mathbf{C}$. This can be efficiently implemented with standard operations.

**Expression of $\nabla L_{\mathbf{q}}$:** The loss on the query frame $L_{\mathrm{q}}$ is formulated in (1c) and in eq. 6 of the main paper, as $L_{\mathrm{q}}(w) = \|r_q(w, f^q)\|^2$, where the residual $r_q$ is defined below:

$$r_q(w, f^q) = R_\theta * \mathbf{C}(w, f^q) \tag{11}$$

Following similar steps than for $L_{\mathrm{r}}$, the gradient $\nabla L_{\mathrm{q}}$ of the loss (1c) w.r.t. the filters $w$ is then computed as,

$$\nabla L_{\mathrm{q}} = 2\left[\frac{\partial r_q(w, f^q)}{\partial w}\right]^{\mathrm{T}} r_q(w, f^q) \tag{12}$$

where $J_q = \frac{\partial r_q(w, f^q)}{\partial w}$ corresponds to the Jacobian of the residual function (11) with respect to the filter map $w$.

$$J_q = R_\theta * \frac{\partial \mathbf{C}(w, f^q)}{\partial w}\,. \tag{13}$$

This leads to the final formulation of the gradient as,

$$\nabla L_{\mathrm{q}} = 2\left[\frac{\partial \mathbf{C}(w, f^q)}{\partial w}\right]^{\mathrm{T}} [R_\theta *]^{\mathrm{T}} r_q(w, f^q)\,. \tag{14}$$

Here, $[R_\theta *]^{\mathrm{T}}$ denotes the transposed convolution with the kernel $R_\theta$.

**Algorithm 1** Filter Predictor module $P$.

---

**Require:** Reference and Query feature maps $f^r$, $f^q$, iterations $N_{\text{iter}}$
1:  $w^0 \leftarrow \texttt{ModelInit}(f^r)$      # Initialize filter map (sec. 3.6 of main paper)
2: **for** $i = 0, \dots, N_{\text{iter}} - 1$ **do**      # Optimizer module loop
3:      $\nabla L(w^n) \leftarrow \texttt{FiltGrad}(w^n, f^r, f^q)$    # Using (10) - (14)
4:      $\alpha_{\text{num}}^n \leftarrow \|\nabla L(w^n)\|^2$
5:      $\alpha_{\text{den}}^n \leftarrow \|J^n \nabla L(w^n)\|^2$      # Apply Jacobian (8) and (13)
6:      $\alpha^n \leftarrow \alpha_{\text{num}}^n / \alpha_{\text{den}}^n$     # Compute step length (3)
7:      $w^{n+1} \leftarrow w^n - \alpha^n \nabla L(w^n)$      # Update filter map
8: **end for**

---

## A.2   Calculation of step-length $\alpha^n$

In this section, we show the calculation of the denominator of $\alpha^n = \frac{\alpha_{\text{num}}^n}{\alpha_{\text{den}}^n}$. The denominator in equation (3) is given by,

$$
\begin{aligned}
\alpha_{\text{den}}^n &= \|J^n \nabla L(w^n)\|^2 \\
&= \left\| J_r(w)|_{w^n} \nabla L(w^n) \right\|^2 + \left\| J_q(w)|_{w^n} \nabla L(w^n) \right\|^2 + \|\lambda_\theta \nabla L(w^n)\|^2
\end{aligned}
\tag{15}
$$

Using equations (8) and (13), we finally obtain:

$$
\begin{aligned}
\alpha_{\text{den}}^n &= \left\| \frac{\partial \sigma_\eta}{\partial \mathbf{C}(w, f^r)} \frac{\partial \mathbf{C}(w, f^r)}{\partial w} \bigg|_{w^n} \nabla L(w^n) \right\|^2 + \left\| R_\theta * \frac{\partial \mathbf{C}(w, f^q)}{\partial w} \bigg|_{w^n} \nabla L(w^n) \right\|^2 + \|\lambda_\theta \nabla L(w^n)\|^2 \\
&= \left\| \frac{\partial \sigma_\eta}{\partial \mathbf{C}(w, f^r)} \mathbf{C}(\nabla L(w^n), f^r) \right\|^2 + \|R_\theta * \mathbf{C}(\nabla L(w^n), f^q)\|^2 + \|\lambda_\theta \nabla L(w^n)\|^2
\end{aligned}
\tag{16}
$$

The relation $\frac{\partial \mathbf{C}(w, f^r)}{\partial w}\big|_{w^n} \nabla L(w^n) = \mathbf{C}(\nabla L(w^n), f^r)$ stems from the linearity of $\mathbf{C}$ in the first argument. All operations in (16) can thus easily be implemented using standard neural network operations. We summarize the different steps taking place within the filter map predictor module $P$ in algorithm 1.

## B   Initial estimate of $w^0$

As explained in Section 3.6 of the main paper, to reduce the number of optimization iterations needed in the filter predictor network $P$, we generate an initial filter map $w^0$, which is then processed by the optimizer module to provide the final discriminative filter $w^* = P(f^r, f^q)$.

We wish that $w^0$ integrates information about the entire reference feature map $f^r$. We thus formulate $w^0$ at location $(i, j)$ as a linear combination of $f_{ij}^r$ and $\bar{f}^r$, where $\bar{f}^r \in \mathbb{R}^D$ is the spatial average reference vector, encoding contextual information. We obtain $w^0$ by solving for the scalar factors $a_{ij}, b_{ij}$ that adhere to the following constraints,

$$
w_{ij}^0 = a_{ij} f_{ij}^r + b_{ij} \bar{f}^r \tag{17a}
$$

$$
(w_{ij}^0)^{\mathrm{T}} f_{ij}^r = \beta \tag{17b}
$$

$$
(w_{ij}^0)^{\mathrm{T}} \bar{f}^r = \gamma \tag{17c}
$$

In the simplest setting, $\beta$ can be set to one and $\gamma$ to zero. However, we let these values be learnt from data. The scalar coefficients $a_{ij}$ and $b_{ij}$ are then easily found by solving these equations, resulting in the following formulation for $w^0$,

$$
w_{ij}^0 = \frac{\left[\beta \|\bar{f}^r\|^2 - \gamma (f_{ij}^r)^{\mathrm{T}} \bar{f}^r\right] f_{ij}^r - \left[\beta (f_{ij}^r)^{\mathrm{T}} \bar{f}^r - \gamma \|f_{ij}^r\|^2\right] \bar{f}^r}{\|\bar{f}^r\|^2 \|f_{ij}^r\|^2 - \left((f_{ij}^r)^{\mathrm{T}} \bar{f}^r\right)^2}
\tag{18}
$$

As already mentioned, $\beta$ and $\gamma$ are learnable weights. In the simplest case, both are just scalars. We call this version **ContextAwareInitializer**. To add further flexibility, they can alternatively be vectors

Figure 1: Plot of the learnt target confidence $y'_\theta$ and weights $v^+_\theta, m_\theta$. The learnt values are shown in red while the initialization of each function is presented in green.

of the same dimension $D$ than the reference feature map $f^r \in \mathbb{R}^{H \times W \times D}$, such that $\beta, \gamma \in \mathbb{R}^D$. We refer to this variant of the initializer as **Flexible-ContextAwareInitializer**. Both versions explicitly integrate context information about the entire reference feature map.

We additionally define a simpler alternative for $w^0$, that we call **SimpleInitializer** for which it is assumed that $(\bar{f}^r_{ij})^\mathrm{T} \bar{f}^r = 0$. As a result, $w_{ij}$ only depends on the reference feature $f^r_{ij}$ at this location, $w^0$ can thus be formulated as:

$$w^0_{ij} = \beta \frac{f^r_{ij}}{\|f^r_{ij}\|} \tag{19}$$

Here, $\beta$ can also be either a scalar (**SimpleInitializer**) or a vector of dimension $D$ (**Flexible-SimpleInitializer**).

In our Global-GOCor module, we use the variant Flexible-ContextAwareInitializer for our initializer module. We defend this choice in our supplementary ablation study Section F. For our Local-GOCor module, we instead use the SimpleInitializer variant of the initializer.

## C   Architecture details

**Expression for $y, v^-, v^+$:**   Here we discuss the parametrization of $y, v^-, v^+$, introduced in the reference loss formulation in Sec. 3.4 of the main paper.

For the implementation, we define $y = v^+ y'$ and $v^- = v^+ m$, with element-wise multiplication. We parametrize $y', v^+, m$ as functions of the distance $d_{ijkl} = \sqrt{(i-k)^2 + (j-l)^2}$ between $w_{ij}$ and the example $f^r_{kl}$, such that $y'_{ijkl} = y'_\theta(d_{ijkl})$, $v^+_{ijkl} = v^+_\theta(d_{ijkl})$, $m_{ijkl} = m_\theta(d_{ijkl})$.

All three are expressed with triangular basis function, as in [2]. For example, the function $y'$ at position $(i, j, k, l)$ is given by:

$$y'_{ijkl} = \sum_{k=0}^{N-1} (y'_\theta)^k \rho_k(d_{ijkl}) \tag{20}$$

with triangular basis functions $\rho_k$, expressed as

$$\rho_k(d) = \begin{cases} \max\left(0, 1 - \frac{|d - k\Delta|}{\Delta}\right), & k < N-1 \\ \max\left(0, \min\left(1, 1 + \frac{d - k\Delta}{\Delta}\right)\right), & k = N-1 \end{cases} \tag{21}$$

We use $N = 10$ basis functions and set the knot displacement to $\Delta = 0.5$ in the resolution of the deep feature space. The final case $k = N-1$ represents all locations $(k, l)$ that are far away from $(i, j)$ and thus can be treated identically.

The coefficients $y'_\theta, v^+_\theta, m_\theta$ are learnt from data, as part of the filter predictor module $P$. For $m$, we constrain the values in the interval $[0, 1]$ by passing the output of (20) through a Sigmoid function. We initialize the target confidence $y'_\theta$ to a Gaussian, with mean equal to 0 and standard deviation equal to 1. The positive weight function $v^+_\theta$ is initialized to a constant so that $v^+_{ijkl} = 1$ while we initialize the function $m_\theta$ with a scaled tanh function.

The initial and learnt values for $y'_\theta, v^+_\theta, m_\theta$ of our Global-GOCor module are visualized in Figure 1. They result from the training of GLU-Net-GOCor on the synthetic *Dynamic* training dataset. We

Figure 2: Visualization of the heat maps corresponding to the learnt target confidence $y$ and weights $v^-, v^+$ for a particular location $(i, j) = (7, 7)$. (a), (b) and (c) show the initialization of each function while (d), (e) and (f) depict the learnt values.

additionally provide the visualization of $y_{ij..}, v_{ij..}^-, v_{ij..}^+ \in \mathbb{R}^{H \times W}$ as heat-maps for a particular location $(i, j)$ in Figure 2.

**Smoothness operator $\mathbf{R}_\theta$:** We now focus on the operator $R_\theta$, introduced in the loss formulation on the query image in Sec. 3.5 of the main paper. $R_\theta \in \mathbb{R}^{K^4 \times Q}$ is a learnable 4D-kernel of spatial size $K$ and $Q$ number of output channels. We set $K = 3$ and $Q = 16$ output channels. For implementation purposes, the 4-D convolution is factorized as two consecutive 2-D convolutional layers, operating over the two first and two latter dimensions respectively. The output dimension of the first 2D-convolution is also set to 16. Note that the kernel $R_\theta$ is learnt, along with all other network parameters, by the SGD-based minimization of the same final network training loss used in the GLU-Net and PWC-Net baselines. This is contrary to the filter map $w$ that is optimized using Eq. 5 and 6 (of the main paper) at each forward pass of the network.

# D   Experimental setup and datasets

In this section, we first provide details about the evaluation datasets and metrics. We then explain the procedure used to create the *Dynamic* dataset, utilized for training state-of-the-art GLU-Net. Finally, we detail the architecture of our baseline network, used for our ablation study, namely BaseNet.

## D.1   Evaluation datasets

**HP:** The HPatches dataset [1] is a benchmark for geometric matching correspondence estimation. It depicts planar scenes, with transformations restricted to homographies. We only employ the 59 sequences labelled with v_X, which have viewpoint changes, thus excluding the ones labelled i_X, which only have illumination changes. Each image sequence contains a query image and 5 reference images taken under increasingly larger viewpoints changes, with sizes ranging from $450 \times 600$ to $1613 \times 1210$.

**ETH3D:** To validate our approach for real 3D scenes, where image transformations are not constrained to simple homographies, we also employ the Multi-view dataset ETH3D [18]. It contains 10 image sequences at $480 \times 752$ or $514 \times 955$ resolution, depicting indoor and outdoor scenes and resulting from the movement of a camera completely unconstrained, used for benchmarking 3D reconstruction. The authors additionally provide a set of sparse geometrically consistent image correspondences (generated by [17]) that have been optimized over the entire image sequence using

the reprojection error. We sample image pairs from each sequence at different intervals to analyze varying magnitude of geometric transformations, and use the provided points as sparse ground truth correspondences. This results in about 500 image pairs in total for each selected interval.

**MegaDepth:** To validate our approach on real scenes depicting extreme viewpoint changes, we use images of the MegaDepth dataset. No real ground-truth correspondences are available, so we use the result of SfM reconstructions to obtain sparse ground-truth correspondences. We follow the same procedure and test images than [19]. More precisely, we use 3D points and project them onto pairs of matching images to obtain correspondences and we randomly sample 1600 pairs of images that shared more than 30 points. It results in approximately 367K correspondences.

**KITTI:** The KITTI dataset [7] is composed of real road sequences captured by a car-mounted stereo camera rig. The KITTI benchmark is targeted for autonomous driving applications and its semi-dense ground truth is collected using LIDAR. The 2012 set only consists of static scenes while the 2015 set is extended to dynamic scenes via human annotations. The later contains large motion, severe illumination changes, and occlusions.

**Sintel:** The Sintel benchmark [3] is created using the open source graphics movie "Sintel" with two passes, clean and final. The final pass contains strong atmospheric effects, motion blur, and camera noise.

## D.2 Evaluation metrics

**AEPE:** AEPE is defined as the Euclidean distance between estimated and ground truth flow fields, averaged over all valid pixels of the reference image.

**PCK:** The Percentage of Correct Keypoints (PCK) is computed as the percentage of correspondences $\tilde{\mathbf{x}}_j$ with an Euclidean distance error $\|\tilde{\mathbf{x}}_j - \mathbf{x}_j\| \leq T$, w.r.t. to the ground truth $\mathbf{x}_j$, that is smaller than a threshold $T$.

**F1:** F1 designates the percentage of outliers averaged over all valid pixels of the dataset [7]. They are defined as follows, where $F_{gt}$ indicates the ground-truth flow field and $F$ the estimated flow by the network.

$$F1 = \frac{\|F - F_{gt}\| > 3 \text{ and } \frac{\|F - F_{gt}\|}{\|F_{gt}\|} > 0.05}{\#\text{valid pixels}} \tag{22}$$

## D.3 Training dataset

In [22], GLUNet is trained on *DPED-ADE-CityScapes*, created by applying synthetic affine, TPS and homography transformations to real images of the DPED [10], CityScapes [5] and ADE-20K [23] datasets. Here, we refer to this dataset as the *Static* training dataset, since it simulates a static scene. While GLU-Net trained on the *Static* dataset obtains state-of-the-art results on geometric matching and optical flow datasets (see Table 1), the *Static* dataset does not capture independently moving objects, present in optical flow data. For this reason, we introduce a *Dynamic* training dataset, created from the original *Static* dataset with additional random independently moving objects. To do so, these objects are sampled from the COCO dataset [12], and inserted on top of the images of the *Static* data using their segmentation masks. To generate motion, we randomly sample affine transformation parameters for the foreground objects, which are independent of the background transformations. This can be interpreted as both the camera and the objects moving independently of each other. The *Dynamic* dataset allows the network to learn the presence of independently moving objects and motion boundaries.

In Table 1, we compare evaluation results of original GLU-Net trained on either the *Static* or the *Dynamic* datasets. While training on the *Dynamic* data leads to worse results on HPatches, it leads to

Table 1: Evaluation results of GLU-Net when trained on the *Static* or the *Dynamic* datasets.

| | HP | | KITTI-2012 | | KITTI-2015 | | Sintel-Cleam | | Sintel-Final | |
|---|---|---|---|---|---|---|---|---|---|---|
| | AEPE ↓ | PCK-5 [%] ↑ | AEPE ↓ | F1 [%] ↓ | AEPE ↓ | F1 [%] ↓ | AEPE ↓ | PCK-5 [%] ↑ | AEPE ↓ | PCK-5 [%] ↑ |
| GLU-Net (*Static*) | **25.05** | **78.54** | 3.34 | **18.93** | 9.79 | 37.52 | 6.03 | 84.21 | 7.01 | 81.92 |
| GLU-Net (*Dynamic*) | 27.01 | 78.37 | **3.14** | 19.76 | **7.49** | **33.83** | **4.25** | **88.40** | **5.50** | **85.10** |

Figure 3: Schematic representation of BaseNet and BaseNet-GOCor, estimating dense flow field w from a pair of reference and query images.

improved performances on all optical flow data, particularly significant on Sintel and KITTI-2015. Only the F1 metric on KITTI-2012 is slightly worse when training on the *Dynamic* dataset. This is consistent with the fact that the *Static* training dataset is in line with HPatches, both restricted to homography transformations, while the *Dynamic* one is better suited for optical flow data, that depict independently moving objects. The *Dynamic* dataset is especially suitable for KITTI-2015 and Sintel, since both represent dynamic scenes, while KITTI-2012 only experiences static 3D scenes. Here, we emphasize that both GLU-Net and GLU-Net-GOCor are trained with exactly the same procedure, introduced in [22].

### D.4 Architecture of BaseNet

We introduce BaseNet, a simpler version of GLU-Net [22], estimating the dense flow fields relating a pair of images. The network is composed of three pyramid levels and it uses VGG-16 [4] as feature extractor backbone. The coarsest level is based on a global correlation layer, followed by a mapping decoder estimating the correspondence map at this resolution. The two next pyramid levels instead rely on local correlation layers. The dense flow field is then estimated with flow decoders, taking as input the correspondence volumes resulting from the local feature correlation layers. Besides, BaseNet is restricted to a pre-determined input resolution $H_L \times W_L = 256 \times 256$ due to its global correlation at the coarsest pyramid level. It estimates a final flow-field at a quarter of the input resolution $H_L \times W_L$, which needs to be upsampled to original image resolution $H \times W$. The mapping and flow decoders have the same architecture as those used for GLU-Net [22].

To create BaseNet-GOCor, we simply replace the global and local correlation layers by respectively our global and local GOCor modules. In the standard BaseNet, the correspondence volume generated by the global correlation layer is passed through a ReLU non linearity [13] and further L2-normalized in the channel dimension, to enhance high correlation values and to down-weight noise values. While beneficial for the standard feature correlation layer, we found the L2-normalization to be slightly harmful for the performance when using our GOCor module. Indeed, our GOCor module inherently already suppresses correlation values at ambiguous matches while enhancing the correct match. We therefore only pass the correspondence volume through a Leaky-ReLU. The rest of the architecture remains unchanged. Schematic representations of BaseNet and BaseNet-GOCor are presented in Figure 3.

Both networks are trained end-to-end, following the same procedure introduced in [22]. We set the batch size to 40 and the initial learning rate of $10^{-3}$, which is further reduced during training.

## E  Additional results

Here, we first look at the impact of the number of Steepest Descent iterations used within the filter predictor module during inference in section E.1. In section E.2, we then give more detailed results for the task of geometric matching. We subsequently provide an extended table of results on optical flow datasets in section E.3 as well as additional results on the ETH3D dataset. We then illustrate the superiority of our approach through multiple qualitative examples in section E.4. Finally, we compare results for different loss parametrization in section E.5.

(a) HPatches

(b) KITTI-2012

(b) Sintel-clean

Figure 4: Evaluation results of GLU-Net-GOCor and PWC-Net-GOCor when increasing the number of steepest descent iterations in either the global or the local GOCor modules. While increasing the number of global iterations, the number of local iterations is fixed to three, and similarly. Note that GLU-Net-GOCor was trained on the *Dynamic* dataset and PWC-Net-GOCor was trained on *3D-Things*. Both networks were trained with three steepest descent iterations for both the local and global GOCor modules, if applicable.

## E.1 Impact of number of inference Steepest Descent iterations

**Impact on performance:** Here, we analyse the impact of the number of Steepest Descent iterations in the global and local GOCor modules used during inference, on the performance of the corresponding network. We first focus on the global GOCor module. In Figure 4, we plot the AEPE and PCK-5px obtained by GLU-Net-GOCor when increasing the number of global steepest descent iterations during inference. Note that the network was trained with three global iterations. On both HPatches and KITTI-2012, GLU-Net-GOCor with three global iterations, i.e. with the same number of iterations than used during training, leads to the best performance. Increasing or decreasing the number of global iterations leads to a significant drop in performance. This is primarily due to our query frame objective (Sec. 3.5 of the main paper), which learns the optimal regularizer weights for the number of steepest descent iterations used during training.

We next look at the impact of the number of steepest descent iterations used in the *local* GOCor module. In Figure 4, we thus plot the AEPE and PCK-5px of GLU-Net-GOCor and PWC-Net-GOCor for different inference number of local optimization iterations. Both networks were trained with three such iterations. Increasing the number of local steepest descent iterations during inference improves the network performances on all datasets. On KITTI-2012 only, the trend is slightly different, however the difference in performance for different number of iterations is insignificant, in the order of 0.01. It is important to note that in the local GOCor module, we only use our robust loss in the reference frame (Sec. 3.4), ignoring the loss on the query frame (Sec. 3.5). Therefore, increasing the number of iterations during inference will in that case make the predicted filter map $w_{ij}$ at location $(i, j)$ more

Table 2: Run time of our method compared to original versions of PWC-Net and GLU-Net, averaged over the 194 image pairs of KITTI-2012. The number of optimization iterations is indicated for the local-GOCor modules. For GLU-Net-GOCor, we use three steepest descent iterations in the global-GOCor.

|  | PWC-Net | PWC-Net-GOCor optim-iter = 3 | PWC-Net-GOCor optim-iter = 7 | GLU-Net | GLU-Net-GOCor optim-iter = 3 | GLU-Net-GOCor optim-iter = 7 |
|---|---|---|---|---|---|---|
| Run-time [ms] | 118.05 | 166.00 | 203.02 | 154.97 | 211.02 | 261.90 |

Figure 5: Visualization of the flow field estimated by GLU-Net and GLU-Net-GOCor for different steepest descent iterations during inference. The image pair is extracted from the clean pass of Sintel. The first number indicates the number of optimization iterations in the global GOCor module while the second refer to the number of steepest descent iterations in the local GOCor module. Both GLU-Net and GLU-Net-GOCor were trained on the *Dynamic* dataset. GLU-Net-GOCor was trained with three steepest descent iterations for both the local and global GOCor modules.

and more discriminative to reference feature $f_{ij}^r$. The final correspondence volume obtained from applying optimized $w^*$ to the query feature map will thus be more accurate.

Taking into consideration solely the performance gain, in the global-GOCor, the best alternative during inference is to use the same number of steepest descent iterations than during training (i.e. three iterations here). For the local-GOCor on the other hand, increasing the number of inference steepest descent iterations leads to better resulting network metrics. However, one must take into account that while increasing the number of inference iterations in the local GOCor module leads to improved performances, it also results in increased inference run-time.

**Impact on run-time:** We thus compare the run-time of our GOCor-networks for different number of optimization iterations in the local GOCor module. For reference, we additionally compare them to their corresponding original networks GLU-Net and PWC-Net. The run-times computed on all images of the KITTI-2012 images are presented in Table 2. The timings have been obtained on the same desktop with an NVIDIA Titan X GPU. All networks output a flow at a quarter resolution of the input images. We up-scale to the image resolution with bilinear interpolation. This up-scaling operation is included in the estimated time.

Therefore, we found that setting seven steepest descent iterations during inference in the local GOCor was a good compromise between excellent performance and reasonable run-time. All results in the main paper are indicated with this setting. Nevertheless, for time-demanding applications, only using three local optimization iterations (i.e. the same number than during training) results in faster GOCor-networks with still a significant performance gain compared to their original feature correlation layer-based networks.

In Figure 5, we visualize the estimated flow field for a pair of Sintel images, by GLU-Net and GLU-Net-GOCor for different optimization iterations. It is very clear that increasing the number of optimization iterations leads to a more accurate estimated flow field. In particular, the estimated flow field becomes more detailed, with well-defined motion boundaries.

### E.2 Additional geometric matching results

Detailed results obtained by GLU-Net and GLU-Net-GOCor on the various view-points of the HP dataset are presented in Table 3. It extends Table 1 of the main paper, that only provides the average over all viewpoint IDs. In addition to the Average End-Point Error (AEPE), we also provide the

Table 3: Details of AEPE and PCK evaluated over each view-point ID of the HPatches dataset. All methods are trained on the *Static* dataset.

| | | I | II | III | IV | V | all |
|---|---|---|---|---|---|---|---|
| GLU-Net | AEPE ↓ | $1.55 \pm 1.80$ | $12.66 \pm 10.43$ | $27.54 \pm 16.05$ | $32.04 \pm 20.01$ | $51.47 \pm 94.77$ | $25.05 \pm 16.67$ |
| | PCK-1px [%] ↑ | 61.72 | 42.43 | 40.57 | 29.47 | 23.55 | 39.55 |
| | PCK-5px [%] ↑ | 96.15 | 84.35 | 79.46 | 73.80 | 58.92 | 78.54 |
| GLU-Net-GOCor | AEPE ↓ | $\mathbf{1.29 \pm 1.31}$ | $\mathbf{10.07 \pm 7.44}$ | $\mathbf{23.86 \pm 14.01}$ | $\mathbf{27.17 \pm 16.84}$ | $\mathbf{38.41 \pm 28.52}$ | $\mathbf{20.16 \pm 13.63}$ |
| | PCK-1px [%] ↑ | **64.93** | **43.86** | **42.52** | **30.68** | **25.78** | **41.55** |
| | PCK-5px [%] ↑ | **96.95** | **86.41** | **82.47** | **76.17** | **65.15** | **81.43** |

Table 4: Results for the optical flow task on the training splits of KITTI [7] and Sintel [3]. A result in parenthesis indicates that the dataset was used for training. PWC-Net* indicates the results stated in the original PWC-Net paper [21]. For all methods, the training dataset is indicated in parenthesis next to the method. When not indicated, the method was trained on *Flying-Chairs* [6] followed by *3D-Things* [11].

| | KITTI-2012 | | KITTI-2015 | | Sintel Clean | | | Sintel Final | | |
|---|---|---|---|---|---|---|---|---|---|---|
| | AEPE ↓ | F1 (%) ↓ | AEPE ↓ | F1 (%) ↓ | AEPE ↓ | PCK-1 (%) ↑ | PCK-5 (%) ↑ | AEPE ↓ | PCK-1 (%) ↑ | PCK-5 (%) ↑ |
| GLU-Net (*Dynamic*) | 3.14 | 19.76 | 7.49 | 33.83 | 4.25 | 62.08 | 88.40 | 5.50 | 57.85 | 85.10 |
| GLU-Net-GOCor (*Dynamic*) (Ours) | **2.68** | **15.43** | **6.68** | **27.57** | **3.80** | **67.12** | **90.41** | **4.90** | **63.38** | **87.69** |
| FlowNet2.0 [11] | 4.09 | - | 10.06 | 30.37 | 2.02 | - | - | 3.14 | - | - |
| SpyNet [15] | 9.12 | - | - | - | 4.12 | - | - | 6.69 | - | - |
| LiteFlowNet [8] | 4.00 | - | 10.39 | 28.50 | 2.48 | - | - | 4.04 | - | - |
| LiteFlowNet2 [9] | 3.42 | - | 8.97 | 25.88 | 2.24 | - | - | 3.78 | - | - |
| PWC-Net* | 4.14 | 21.38 | 10.35 | 33.67 | 2.55 | - | - | 3.93 | - | - |
| PWC-Net (*ft 3D-Things*) | 4.34 | 20.90 | 10.81 | 32.75 | 2.43 | 81.28 | 93.74 | 3.77 | 76.53 | 90.87 |
| PWC-Net-GOCor (*ft 3D-Things*) (Ours) | **4.12** | **19.31** | **10.33** | **30.53** | **2.38** | **82.17** | **94.13** | **3.70** | **77.34** | **91.20** |
| FlowNet2 (*ft Sintel*) | 3.54 | - | 9.94 | 28.02 | (1.45) | - | - | (2.19) | - | - |
| PWC-Net* (*ft Sintel*) | 2.94 | 12.70 | 8.15 | 24.35 | (1.70) | - | - | (2.21) | - | - |
| PWC-Net (*ft Sintel*) | 2.87 | 11.97 | 8.68 | 23.82 | (1.76) | (87.24) | (95.37) | (2.23) | (83.61) | (93.61) |
| PWC-Net-GOCor (*ft Sintel*) (Ours) | **2.60** | **9.67** | **7.64** | **20.93** | (1.74) | (87.93) | (95.54) | (2.28) | (84.15) | (93.71) |

standard deviation over the End-Point Error per image. It represents the distribution of EPE per image, averaged over all images of each view-point. Note that increasing view-point IDs lead to increasing geometric transformations due to larger changes in viewpoint.

Our approach GLU-Net-GOCor outperforms original GLU-Net for each viewpoint ID. Particularly, GLU-Net-GOCor is significantly more robust for large view-point changes, such as those experienced in Viewpoint V, with an AEPE of 38.41 against 51.47 for original GLU-Net. Besides, GLU-Net-GOCor always obtains a narrower distribution of errors. This implies that our approach enables the network to have a more steady performance over the whole dataset.

## E.3 Additional optical flow results

**Extended optical flow results:** Table 4 here extends Table 3 of the main paper with more results on optical flow datasets. Specifically, we compare our approaches with other state-of-the-art networks applied to the train splits of the KITTI and Sintel datasets. Similarly to PWC-Net, these other methods, such as LiteFlowNet [8], rely on local correlation layers at multiple levels to infer the final flow field relating a pair of images. Our local GOCor module could therefore easily be integrated into any of these networks in place of the local correlation layers.

In Table 4, PWC-Net* refers to the results presented in the original PWC-Net publication [21]. In the middle section, we show the evaluation results of PWC-Net*, as well as PWC-Net-GOCor and PWC-Net both further finetuned on *3D-Things* according to the same schedule. In the last section of the table, we focus on the PWC-Net variants finetuned on the *Sintel* training dataset. For a fair comparison, we here also provide the official PWC-Net* ft Sintel results, as well as the PWC-Net-GOCor and PWC-Net versions that we finetuned on *Sintel*. Our PWC-Net-GOCor outperforms both PWC-Net* and PWC-Net on the KITTI data by a large margin, while obtaining similar results on the training set of Sintel. As already mentioned in Sec. 4.2 of the main paper, this highlights the generalization capabilities of our GOCor module.

Table 5: Detailed results on the test set of Sintel benchmark for different regions, velocities (s), and distances from motion boundaries (d). All methods are trained on *Flying-Chairs* [6] followed by *3D-Things* [11], and further finetuned on the training split of *Sintel*.

| | Sintel-Clean | | | | | | | | |
|---|---|---|---|---|---|---|---|---|---|
| | EPE-all | EPE matched | EPE unmatched | d0-10 | d10-60 | d60-140 | s0-10 | s10-40 | s40+ |
| PWC-Net* (*ft-Sintel*) | 4.386 | 1.719 | 26.166 | 4.282 | 1.657 | **0.657** | **0.606** | 0.2070 | 28.783 |
| PWC-Net (*ft-Sintel*) | 4.637 | 1.951 | 26.571 | 4.018 | 1.626 | 1.040 | 0.649 | 2.070 | 30.671 |
| PWC-Net-GOCor (*ft-Sintel*) | **4.195** | **1.660** | **24.909** | **3.843** | **1.448** | 0.778 | 0.609 | **1.914** | **27.552** |

| | Sintel-Final | | | | | | | | |
|---|---|---|---|---|---|---|---|---|---|
| | EPE-all | EPE matched | EPE unmatched | d0-10 | d10-60 | d60-140 | s0-10 | s10-40 | s40+ |
| PWC-Net* (*ft-Sintel*) | **5.042** | **2.445** | **26.221** | 4.636 | 2.087 | **1.475** | 0.799 | 2.986 | **31.070** |
| PWC-Net (*ft-Sintel*) | 5.300 | 2.576 | 27.528 | 4.717 | 2.204 | **1.580** | 0.929 | 2.994 | 32.584 |
| PWC-Net-GOCor (*ft-Sintel*) | **5.133** | **2.458** | **26.945** | **4.504** | **2.063** | 1.603 | **0.834** | **2.906** | 31.858 |

Here, we also present the evaluation results on the test set of the Sintel dataset in Table 5. We compare PWC-Net and PWC-Net-GOCor, both finetuned on the training split of *Sintel*. For reference and as previously, we also present the official results from the PWC-Net publication [21], denoted as PWC-Net*. PWC-Net-GOCor outperforms both PWC-Net and PWC-Net* on the clean pass. On the final pass, PWC-Net-GOCor obtains better performance than PWC-Net, but slightly worse results than PWC-Net*. The authors of PWC-Net employ special data augmentation strategies and training procedures for fine-tuning, which are not shared as PyTorch code by the authors. The finetuning procedure that we employed for our PWC-Net-GOCor and standard PWC-Net is therefore different from the one used for in the official PWC-Net results (PWC-Net*). However, we finetuned both PWC-Net and PWC-Net-GOCor with the same setting and procedure, enabling fair comparison between the two. PWC-Net-GOCor performs particularly better in regions with large motions and close to the motion boundaries. This is in line with the behavior of the GOCor module observed previously, according to which the GOCor module particularly improves performance on large displacements.

**Performance on occlusion data:** Here, we focus specifically on the performance of GOCor in occluded regions. As shown in Tab. 5, in occluded regions ("EPE unmatched") of the Sintel test set, GOCor provides relative improvements of 6.25% and 2.16% on the clean and final pass respectively. In Tab. 6, we present the

Table 6: AEPE/F1 [%] on KITTI-2015.

|  | Not occluded | Occluded | All |
|---|---|---|---|
| GLU-Net | 4.67 / 27.83 | 21.95 / 67.44 | 7.49 / 33.83 |
| GLU-Net-GOCor | **4.22 / 22.03** | **19.07 / 58.61** | **6.68 / 27.57** |
| PWC-Net | 5.40 / 25.16 | 34.39 / 78.58 | 10.81 / 32.75 |
| PWC-Net-GOCor | **5.02 / 23.53** | **34.06 / 77.84** | **10.33 / 30.53** |

details of the metrics on occluded and non-occluded regions of KITTI-2015. GOCor improves the performance of PWC-Net and GLU-Net in occluded regions of a substantial amount as compared to the feature correlation layer.

**Additional results on ETH3D:** For completeness, in Figure 6, we also present the evaluation results of PWC-Net and PWC-Net-GOCor applied to the ETH3D images, sampled at increasingly high intervals. Indeed, for small intervals, finding correspondences strongly resembles optical flow task while increasing the interval leads to larger displacements. Optical flow network PWC-Net can thus be very suitable, particularly when estimating the flow at small intervals. In Figure 6, both PWC-Net and PWC-Net-GOCor are finetuned on *3D-Things* according to the same schedule. For reference, we additionally provide the results of PWC-Net*, which refers to the official pre-trained weights provided by the authors, after training on *Flying-chairs* and *3D-Things*. We also provide the results of PWC-Net*, PWC-Net and PWC-Net-GOCor further finetuned on *Sintel*.

For all intervals, and independently of the dataset it was trained on, our approach PWC-Net-GOCor obtains better metrics than both PWC-Net and PWC-Net*. Particularly, the gap in performance broadens with the intervals between the frames, implying that our module is especially better at handling large view-point changes. This particular robustness to large displacements was similarly observed when our GOCor modules were integrated into GLU-Net (Sec. 4.1). It highlights that the improved performances brought by our GOCor approach compared to the feature correlation layer *generalize to different networks*.

Figure 6: Quantitative results on ETH3D [18] images. AEPE, PCK-1 and PCK-5 are computed on pairs of images sampled from consecutive images of ETH3D at different intervals. PWC-Net* refers to the official pre-trained weights provided by the authors.

### E.4 Qualitative examples

Here, we first present qualitative comparisons of PWC-Net and PWC-Net-GOCor. In Figure 8, we show examples of PWC-Net and PWC-Net-GOCor applied to images of optical flow datasets KITTI-2012 and KITTI-2015. Both networks are trained on *3D-Things*. PWC-Net-GOCor shows more defined motion boundaries and generally more accurate estimated flow fields. Similarly, we present examples on the clean pass of the Sintel training set in Figure 9. PWC-Net-GOCor captures more detailed flow fields. This is for instance illustrated in the first example of Figure 9, where PWC-Net-GOCor correctly identified the foot contrary to PWC-Net, which failed to capture it. However, both PWC-Net and PWC-Net-GOCor may fail on small and rapidly moving objects, such as the arm in the last example of Figure 9.

In Figure 10, we additionally present qualitative results on the KITTI images when the networks are finetuned on Sintel. Indeed, in Table 4, we showed that while both PWC-Net and PWC-Net-GOCor obtain very similar results on training data *Sintel*, PWC-Net-GOCor performs largely better on the KITTI datasets compared to original PWC-Net, especially in terms of F1 metric. PWC-Net-GOCor also obtains visually more accurate flow fields on the KITTI images.

Besides, we show the advantage of our approach as compared to the feature correlation layer when integrated in GLU-Net. In Figure 11, we visually compare GLU-Net and GLU-Net-GOCor when applied to images of the clean pass of the Sintel benchmark and to images of the ETH3D dataset. In the case of the ETH3D images, the pairs of images are taken by two different cameras simultaneously. The camera of the first images has a field-of-view of 54 degrees while the other camera has a field of view of 83 degrees. They capture images at a resolution of $480 \times 752$ or $514 \times 955$ depending on the scenes and on the camera. The exposure settings of the cameras are set to automatic, allowing the device to adapt to illumination changes. On the Sintel images, GLU-Net-GOCor achieves sharper object boundaries and generally more correct estimated flow fields compared to original GLU-Net. On the ETH3D images, our approach is more robust to illumination changes and light artifacts, leading to better visual outputs.

Finally, we qualitatively compare the output of GLU-Net-GOCor and GLU-Net on example pairs of the MegaDepth dataset in Figure 7. It is obvious that GOCor provides an increased robustness to very large geometric view-point changes, such as large scaling or perspective variations.

### E.5 Results for smooth version of the reference loss

In Section 3.4 of the main paper, we defined our robust and learnable objective function for integrating reference frame information as $L_{\mathrm{r}}(w; f^r, \theta) = \left\| \sigma_\eta \big( \mathbf{C}(w, f^r); v^+, v^- \big) - y \right\|^2$ (eq. 5 of main paper), with $\sigma_\eta(\mathbf{C}(w, f^r); v^+, v^-) = \frac{v^+ - v^-}{2} \left( \sqrt{\mathbf{C}(w, f^r)^2 + \eta^2} - \eta \right) + \frac{v^+ + v^-}{2} \mathbf{C}(w, f^r)$ (eq. 4 of main paper).

Here, setting $\eta > 0$ enables to avoid the discontinuity in the derivative of $\sigma$ at $\varepsilon = 0$. We analyze two different settings for $\eta$ when integrated in GLU-Net-GOCor. Specifically, in Table 7, we compare the results obtained by GLU-Net-GOCor on optical flow data, when setting $\eta = 0$ or $\eta = 0.1$. Both values obtain very similar results. Therefore, for simplicity and efficiency, we use $\eta = 0$ in all other experiments.

Table 7: Comparison of different parametrisation for our robust loss formulation $L^r$. Both GLU-Net-GOCor are trained on the *Dynamic* dataset with three optimization iterations and evaluated with three and seven iterations for respectively the global-GOCor and the local-GOCor modules.

| | KITTI-2012 | | KITTI-2015 | | Sintel Clean | | | Sintel Final | | |
|---|---|---|---|---|---|---|---|---|---|---|
| | AEPE $\downarrow$ | F1 [%] $\downarrow$ | AEPE $\downarrow$ | F1 [%] $\downarrow$ | AEPE $\downarrow$ | PCK-1 [%] $\uparrow$ | PCK-5 [%] $\uparrow$ | AEPE $\downarrow$ | PCK-1 [%] $\uparrow$ | PCK-5 [%] $\uparrow$ |
| GLU-Net | 3.14 | 19.76 | 7.49 | 33.83 | 4.25 | 62.08 | 88.40 | 5.50 | 57.85 | 85.10 |
| GLU-Net-GOCor, $\eta = 0$ | 2.68 | 15.43 | 6.68 | 27.57 | 3.80 | 67.12 | 90.41 | 4.90 | 63.38 | 87.69 |
| GLU-Net-GOCor, $\eta = 0.1$ | 2.70 | 15.51 | 6.92 | 28.06 | 3.74 | 66.72 | 90.42 | 4.81 | 63.04 | 87.69 |

(A)

| Query image | Reference image | BaseNet (**I**) |

| BaseNet + Global-GOCor $L_r$ (**III**) | BaseNet + Global-GOCor $L_r$+ $L_q$ (**IV**) | BaseNet + Global-GOCor $L_r$+ $L_q$ + Local-GOCor $L_r$ (**V**) |

(B)

| Query image | Reference image | BaseNet (**I**) |

| BaseNet + Global-GOCor $L_r$ (**III**) | BaseNet + Global-GOCor $L_r$+ $L_q$ (**IV**) | BaseNet + Global-GOCor $L_r$+ $L_q$ + Local-GOCor $L_r$ (**V**) |

Figure 12: Qualitative analysis of the different components of our approach, when the corresponding networks are applied to images of the ETH3D dataset. All models are trained on the *Dynamic* dataset. We visualize the query images warped according to the flow fields estimated by the networks. The warped query images should resemble the reference images.

(A) KITTI-2015

Reference image  Ground-truth  BaseNet (**I**)

BaseNet + Global-GOCor $L_r$ (**III**)  BaseNet + Global-GOCor $L_r$ + $L_q$ (**IV**)  BaseNet + Global-GOCor $L_r$ + $L_q$ + Local-GOCor $L_r$ (**V**)

(B) Sintel-clean

Reference image  Ground-truth  BaseNet (**I**)

BaseNet + Global-GOCor $L_r$ (**III**)  BaseNet + Global-GOCor $L_r$ + $L_q$ (**IV**)  BaseNet + Global-GOCor $L_r$ + $L_q$ + Local-GOCor $L_r$ (**V**)

Figure 13: Qualitative analysis of the different components of our approach, when the corresponding networks are applied to images of the Sintel dataset, clean pass. All models are trained on the *Dynamic* dataset. We plot directly the estimated flow field for each image pair.

## F  Additional ablation study

For completeness, in Table 8, we provide a similar ablation study as that of the main paper Sec. 4.5, when BaseNet is trained on the *Static* data, instead of the *Dynamic* one. The same conclusions apply. We thus perform the additional ablative experiments when training all BaseNet variants on the *Static* data.

**Qualitative ablation study:**  We visualize the quality of the estimated flow fields outputted by BaseNet (**I**), BaseNet + Global-GOCor $L_r$ (**III**), BaseNet + Global-GOCor $L_r$ + $L_q$ (**IV**) and BaseNet + Global-GOCor $L_r$ + $L_q$ + Local-GOCor $L_r$ (**V**) when applied to images of the ETH3D dataset and of optical flow datasets Sintel and KITTI in respectively Figures 12 and 13. Example (A) of Figure 12 shows the benefit of our global GOCor module as compared to the global feature correlation layer. Indeed, BaseNet does not manage to correctly capture the geometric transformation between the query and the reference images. On the other hand, thanks to our global GOCor module, BaseNet + Global-GOCor estimates a much more accurate transformation relating the frames. In example (B), we illustrate the gradual improvement from version (III) to (V). While introducing Global-GOCor with only the reference loss $L_r$ (version III) makes the estimated flow more stable than BaseNet, especially in the background, the representation of the slide object on the warped image is still shaky. Adding the query loss $L_q$ (version IV) smooths the estimated flow, and therefore the warped query image according to this flow. The later looks visually much better because of additional smoothness. Finally, further substituting the local feature correlation layers with our local GOCor module (version V) finishes to polish the result. The slide object in that case looks almost perfect and artifacts in the background are partially removed.

The impact of the query frame objective $L_q$ in our global GOCor module is further illustrated in example (A) of Figure 13. Introducing $L_q$ enables to smooth the estimated flow field and to remove part of the artifacts. Finally, the advantage of our local GOCor module as opposed to the local feature correlation layer is visualized in both examples of Figure 13. From version (IV) to (V), the local

Table 8: Ablation study. All networks are trained on the *Static* dataset. The GOCor modules are trained and evaluated with 3 steepest descent iterations.

|  |  | HP | | KITTI-2012 | | KITTI-2015 | |
|---|---|---|---|---|---|---|---|
|  |  | AEPE $\downarrow$ | PCK-5 [%] $\uparrow$ | AEPE $\downarrow$ | F1 [%] $\downarrow$ | AEPE $\downarrow$ | F1 [%] $\downarrow$ |
| I | BaseNet | 26.73 | 65.30 | 4.95 | 42.49 | 11.52 | 61.90 |
| II | BaseNet + NC-Net | 24.59 | 66.62 | 5.00 | 39.80 | 12.44 | 62.96 |
| III | BaseNet + Global-GOCor $L_r$ | 22.80 | 70.00 | 4.43 | 34.81 | 10.93 | 55.73 |
| IV | BaseNet + Global-GOCor $L_r + L_q$ | 22.16 | 70.54 | 4.36 | 34.15 | 10.97 | 55.62 |
| V | BaseNet + Global-GOCor $L_r + L_q$ + Local-GOCor $L_r$ | 22.00 | 74.80 | **4.02** | **31.24** | **9.92** | **50.54** |
| VI | BaseNet + Global-GOCor $L_r + L_q$ + Local-GOCor $L_r + L_q$ | **21.96** | **75.26** | 4.24 | 33.43 | 10.20 | 53.53 |

GOCor module can recover sharper motion boundaries and the estimated flow is generally more accurate. Moreover, remaining artifacts in the background are removed.

**Comparison with post-processing method NC-Net:**  Here, we investigate the impact of post-processing method NC-Net [16] and show comparisons to BaseNet and our approach in Table 8. When trained on the *Static* dataset, including post-processing module NC-Net following the global correlation layer (version **II**) leads to better results than original version BaseNet (**I**) on the HPatches and the KITTI-2012 datasets. However, on the KITTI-2015 images, adding NC-Net results in worse performance. This is due to the fact that NC-Net uses correspondences with high confidences to support other uncertain neighboring matches. However, in the case of independently moving objects, neighboring matches can correspond to completely different motions, which breaks the assumption of the neighborhood consensus constraint used in NC-Net. Since it cannot cope with independently moving objects, NC-Net obtains worse results on KITTI-2015, which depicts dynamic scenes. This is contrary to HPatches which present planar scenes with homographies and to KITTI-2012, which is restricted to static scenes. This observation is also emphasized by the ablation study in Table 3 of the main paper, where BaseNet + NC-Net is trained on the *Dynamic* dataset. In that case, the performance of the resulting network is much worse than original BaseNet on all datasets. This is again due to the inability of NC-Net to handle moving objects, in that case, present in the training dataset. This shows the advantage of our method, which instead of applying 4D convolutions to post-process the correspondence volume, integrates them *before* the correlation operation itself.

**Impact of objective function in the Local GOCor:**  In Table 8, we analyse the impact of both terms $L_r, L_q$ of our objective function $L$ when used in our Local GOCor module. Comparing versions (**V**) and (**VI**), we found that adding the loss on the query frame $L_q$ (Sec. 3.5) for Local-GOCor is harmful for its performance, particularly on the KITTI-datasets. Besides, adding the regularizer loss in the local GOCor level leads to longer training and inference run times. We therefore do not include it, our best version of BaseNet-GOCor resulting in (**V**).

**Impact of number of training optimization iterations:**  Here, we investigate the influence of the number of training steepest descent iterations. We train multiple BaseNet-GOCor networks, gradually increasing the number of training optimization iterations within the global and the local GOCor modules. All networks are trained on the *Static* dataset. We evaluate all variants on the HPatches, KITTI and Sintel datasets and present the results in Table 9. We use the same number of optimization iteration during evaluation as during training. We additionally measure the inference run-time of each network, computed as the average over the 194 KITTI-2012 images on an NVIDIA Titan X GPU.

Training and evaluating with more steepest descent iterations consistently leads to better performances on all metrics and all datasets. It is also interesting to note that BaseNet + Global-GOCor without going through the optimizer (0 iteration) already outperforms the original BaseNet. In that case, the improvement is solely due to our powerful initialization $w^0$.

Nevertheless, the improvement in performance when increasing the number of optimization iterations comes at the expense of inference and training time. As a result, we trained all our GOCor modules with three iterations which presented a satisfactory trade-off between inference time and accuracy. Besides, it must also be noted that for time-critical applications, using a single optimization iteration for both local and global GOCor modules already leads to significant improvements over the standard feature correlation layer.

**Impact of performing global correlation with interchanging query and reference features:**  We also experimented with interchanging the query and reference frames at the global level, and then fusing the two resulting GOCor correspondence volumes before passing them to the flow estimation decoder. However, we only observed marginal improvements. e.g., on KITTI-2015 it obtains an EPE of 11.07 and an F1 of 54.68% compared to 10.97 EPE and 55.62% F1 for the baseline BaseNet

Table 9: Analysis of the number of training optimization iterations. Both Local-GOCor and Global-GOCor layers are trained and evaluated with the same number of iterations. All networks are trained on the *Static* dataset.

| | KITTI-2012 | | | HP | | | Sintel-clean | | |
|---|---|---|---|---|---|---|---|---|---|
| | Run-time [ms] | AEPE↓ | F1 [%]↓ | AEPE↓ | PCK-1 [%]↑ | PCK-5 [%]↑ | AEPE↓ | PCK-1 [%]↑ | PCK-5 [%]↑ |
| BaseNet | 63.20 | 4.95 | 42.49 | 26.73 | 12.02 | 65.30 | 7.78 | 12.46 | 74.52 |
| BaseNet-GOCor, optim-iter = 0 | 66.80 | 4.87 | 37.42 | 26.47 | 16.40 | 66.42 | 6.94 | 27.63 | 77.20 |
| BaseNet-GOCor, optim-iter = 1 | 70.52 | 4.18 | 32.95 | 21.91 | 20.73 | 72.82 | 6.49 | 31.03 | 79.87 |
| BaseNet-GOCor, optim-iter = 3 | 82.42 | 4.02 | 31.24 | 22.00 | 23.68 | 74.80 | 6.32 | 33.81 | 80.72 |
| BaseNet-GOCor, optim-iter = 5 | 94.85 | 3.83 | 29.14 | 20.68 | 25.99 | 76.88 | 6.34 | 38.10 | 80.91 |

Table 10: Analysis of the impact of the initializer for the Global-GOCor. All networks are trained on the *Static* with three optimization iterations. They are evaluated with the same number of iterations.

| | HP | | | KITTI-2012 | | Sintel-clean | | |
|---|---|---|---|---|---|---|---|---|
| | AEPE↓ | PCK-1 [%]↑ | PCK-5 [%]↑ | AEPE↓ | F1 [%]↓ | AEPE↓ | PCK-1 [%]↑ | PCK-5 [%]↑ |
| BaseNet + Global-GOCor, ZeroInitializer | 23.22 | 14.05 | 67.53 | 4.77 | 38.67 | 7.56 | 14.94 | 74.96 |
| BaseNet + Global-GOCor, SimpleInitializer | 24.69 | 13.17 | 66.82 | 4.81 | 37.86 | 7.32 | 20.31 | 76.12 |
| BaseNet + Global-GOCor, Flexible-SimpleInitializer | 25.53 | 12.82 | 66.97 | 4.83 | 38.71 | 7.30 | 20.48 | 76.24 |
| BaseNet + Global-GOCor, ContextAwareInitializer | 25.13 | 15.01 | 67.14 | 4.87 | 39.14 | 7.43 | 19.19 | 75.60 |
| BaseNet + Global-GOCor, Flexible-ContextAwareInitializer | **22.16** | **17.03** | **70.54** | **4.36** | **34.15** | **7.21** | **20.68** | **77.09** |

+ Global-GOCor $L_r + L_q$ (IV). Considering that computing the GOCor correspondence volume twice increases the inference time of the model and the limited improvement of performance brought by fusing the two resulting correspondence volumes, we did not include this alternative in the final model.

**Impact of filter initializer** $w^0$**:** As detailed in Sec. B, we introduced several versions of our filter initializer module. Here, we train BaseNet with standard local feature correlation layers and our Global-GOCor module (using both our loss on the reference and on the query images) integrated in place of the global correlation layer. We experiment with different variants of the initializer and present the corresponding evaluation results in Table 10. The version ZeroInitializer initializes $w^0$ to a zero tensor. Compared to all others, this initialization lacks accuracy for the final network, particularly on the optical flow datasets. In the SimpleInitializer versions which do not include global context information, adding more flexibility in the form of a learnt vector does not seem to help. However, in the case where context information is included, compared to ContextAwareInitializer, the Flexible variant significantly gains from increased flexibility. Initializer module Flexible-ContextAwareInitializer appears to be the best alternative for our Global-GOCor module.

Query image&emsp;&emsp;&emsp;Reference image&emsp;&emsp;&emsp;GLU-Net&emsp;&emsp;&emsp;GLU-Net-GOCOr

Figure 7: Qualitative examples of GLU-Net and GLU-Net-GOCor applied to images of the MegaDepth dataset. Both models are trained on the *Dynamic* training data.

Figure 8: Qualitative examples of PWC-Net and PWC-Net-GOCor applied to images of KITTI-2012 and KITTI-2015. Both models are trained on *Flying-Chairs* followed by *3D-Things*.

Figure 9: Qualitative examples of PWC-Net and PWC-Net-GOCor applied to images of Sintel-clean. Both models are trained on *Flying-Chairs* followed by *3D-Things*.

Figure 10: Qualitative examples of PWC-Net and PWC-Net-GOCor applied to images of KITTI2012 and 2015. Both models are finetuned on *Sintel*.

(a) ETH3D images

Query image        Reference image        GLU-Net        GLU-Net-GOCOr

(b) Sintel-clean images

Figure 11: Qualitative examples of GLU-Net and GLU-Net-GOCor applied to images of (a) the ETH3D dataset and (b) the Sintel dataset, Clean pass. Both models are trained on the *Dynamic* dataset. In the case of the ETH3D images, we visualize the query images warped according to the flow fields estimated by the network. The warped query images should resemble the reference images. In the case of Sintel images, we plot directly the estimated flow field for each image pair.