[Reviews · NeurIPS 2020]

Review 1

Summary and Contributions: The authors introduce a module that refines the feature map before computing a correlation volume for dense correspondence problem. Iteratively updating the feature map of the reference image with learned matching priors from the reference image itself and query image enables robust correspondence estimation. Extensive experiments are conducted on three datasets; HPatches, KITTI, sintel.

Strengths: - I like the idea of refining a feature map before computing correlation volume that has not been heavily explored before. - The interpretation of GOCor module with optimization-based meta-learning techniques is interesting. - The generalization ability of GOCor module makes it can be integrated with any dense correspondence estimation algorithms. - The above strengths are demonstrated through extensive experiments and thoughtful ablation studies. - The paper is generally well written and organized with promising results

Weaknesses: - Regarding to the objectives described in Sec. 3.4. and 3.5, the intuitions of them would be more solid if similar attempts are referred. - Imposing constraint of Sec. 3.4 (i.e. w^t_ij*f^r_kl =0 when (k, l) != (i, j)) increase discriminability power of f^r_ij since it is encouraged to be far away from all f^r_kl. However, this may come with the cost of the robustness to variations making w^t sensitive to large appearance or geometric variations. The smoothness constraint of Sec. 3.5 (or from the pyramidal structure of GLU-net and PWC-net) may compensate for this trade-off, but this is somewhat heuristic. The experiments are conducted on classical dense correspondence problems where the variations might be mild, so additional experiments on semantic matching task [23,38,39,40] would be expected where larger appearance and geometric variations exist than traditional stereo or optical flow estimation tasks. The results when combined with other matching algorithms that do not impose smoothness constraint via coarse-to-fine scheme would be interesting.

Correctness: See Weaknesses section.

Clarity: The paper is generally well written. In my opinion, Sec. 3.2 has some repetitive descriptions overall Sec. 3 and can be reduced. The spared room may include the ablation study of Sec. E.1 and E. 2 in supplementary material that I found useful.

Relation to Prior Work: The Related Work section is well-written. Authors might want to cite these as related works: [A] Dynamic Filter Networks, NeurIPS'2016 - where filters are generated dynamically conditioned on an input [B] SuperGlue: Learning Feature Matching with Graph Neural Networks, CVPR'2020 - where the features are updated with attentional graph neural networks whose edges are defined within the same image (intra-image edges) or the other image (inter-image edges).

Reproducibility: Yes

Additional Feedback: Regarding to the objective function of eq. (5), what about cross entropy rather than least square as proposed in [23]? It is reported that cross entropy loss function enable the features to be more discriminative than L2 distance. -- After rebuttal -- I appreciate the authors' response to my concerns and upgrade my original recommendation as "A good submission (7)".


Review 2

Summary and Contributions: This paper presents globally optimized correspondence volumes (GOCor), which is a fully differentiable dense matching module, to address the limitations of previous feature correlation layer. The previous feature correlation layer is formulated to measure the similarity between each pixel from source and target images (or reference and query images) independently, and did not leverage the priors on the images, thus providing the limited capability. Unlike this, the proposed method is capable of effectively learning spatial matching priors to resolve further matching ambiguities. By introducing the filter predictor on the feature of a source image, the extracted correlation volumes with GOCor provide more effective encoding of the matching confidence. Experiments on geometric matching and optical flow have shown the robustness of GOCor compared to existing methods.

Strengths: - The problem formulation that catches the limitation of raw matching cost makes sense and would be a very important further direction, because the feature correlation layer has been popularly used in many applications such as semantic matching, video object segmentation, and few-shot segmentation. - The motivation that reference and query images have a useful prior for dense matching also makes sense. - Two tailored loss functions for reference frame and query frame are well designed. - Thorough experiments on geometric matching and optical flow, as well as ablation study, prove the superiority of the proposed method. - Nice visualizations in Fig. 1, 2, 3 would help the reader to understand this paper well. - This paper is well organized and written.

Weaknesses: There is no strong weakness, but here are some minor issues. - Why the filter predictor P should be built after CNN feature of a source image should be discussed. Ideally, the network module could be applied to not only source but also target images, or target image only. Such an ablation study make an architectural design much stronger. - Is the optimizer described in L231-242 a offline process or online process? - There is a lack of computational complexity.

Correctness: Correct.

Clarity: This paper is well organized and written.

Relation to Prior Work: Yes

Reproducibility: Yes

Additional Feedback: I strongly agree other reviewer's rating, so I will keep my initial rating, "acceptance".


Review 3

Summary and Contributions: This paper describes a neural network module that computes match confidence values for the prediction of dense correspondences between two images. In current networks, the basic information for this task is computed by a so-called correspondence volume (or feature correlation) layer. Such a layer computes a measure of similarity between deep features at corresponding image locations for every location of the reference image and for every vector in a predefined, finite set of image displacements (or, in the global approach, for all possible integer-values displacements). In contrast, the proposed method uses information that a correspondence volume layer does not have access to, namely, (i) the distribution of deep features for image locations with similar appearance and (ii) prior knowledge about properties of the correspondence field, and specifically its smoothness and the uniqueness of the correspondence for each reference location.

Strengths: The correspondence confidence values computed by the proposed method optimize a global similarity criterion, rather than a local one, and achieves superior performance as a result. This is achieved through minimizing a simple and well-motivated loss function. The idea (lines 202-207) of imposing the smoothness constraint on the matching confidence values, as opposed to the displacement field, is interesting and novel. In addition, (the convolution kernel of) this smoothness regularizer is learned during training, rather than being hand-crafted, and can therefore potentially match the data better. The module generalizes well to domains characterized by new image content and motion patterns. The module can be plugged into standard neural networks to replace a correspondence volume layer. Experiments demonstrate superior performance, by significant margins, when the proposed module replaces correspondence volume modules in recent state-of-the-art networks for geometric correspondences and optical flow. For flow, the improvement margins are particularly good on natural scenes (as opposed to synthetic animations), even when training is done on animations. This speaks to the ability of the proposed system to generalize to different domains, a point emphasized also by one of the ablation studies (compare entries II and III in Table 3, page 8).

Weaknesses: The advantages mentioned above are achieved at the cost of an optimization procedure that is performed at inference time. However, a loop-unrolling scheme is proposed to make this optimization efficient, albeit approximate, so this aspect is not a major drawback. Some claims are made in the description of the method that, while plausible, are not evaluated experimentally. Specifically, the proposed query-frame objective does not enforce uniqueness, but merely provides flexibility for the regularizer potentially to learn that correspondences are unique. It is not clear whether this makes things better or worse, and a focused study of this point would be useful. In the terminology of the paper, does the regularizer indeed learn “peak-enhancing operators?” Another useful ablation study would show that computing smoothness on confidence values is better than smoothness on displacement values. This is an interesting and plausible point that deserves empirical support. Similar considerations hold for a more detailed analysis of what happens in the presence of occlusions. While no claim is made about this point, it would be interesting at least to show some anecdotal examples of an optical flow map near occlusion boundaries: To what extent does smoothness regularization blur the flow map? Some of the supplementary materials help to some extent in some of these directions.

Correctness: The claims and the mathematics are correct. The experimental methodology is also correct.

Clarity: The paper is very well written, with good structure, well-designed mathematical notation, good English, and clear pictures.

Relation to Prior Work: Yes. See discussion above.

Reproducibility: Yes

Additional Feedback: The “weaknesses” listed above are really just a plea for more insights about an interesting method. I am fully aware of the difficulty of adding material in an 8-page paper, so these further insights may have to be left for future publications, or for the supplementary materials. I would report running times in the main paper, rather than in the supplementary materials, given that inference involves optimization. The authors' responses to my technical remarks are well taken, and strengthen my original assessment that this is a good paper.


Review 4

Summary and Contributions: This paper proposes a novel cross-correlation layer called GOCor that explicitly takes into consideration the underlying structure of the reference and query images and the regularity exhibited in the scene. The proposed layer is well motivated in the sense that the standard feature correlation layer does not disambiguate between multiple similar regions. And the proposed one can probably alleviate these issues by an internal optimization procedure that explicitly accounts for similar ones. Improvements have been demonstrated in both geometric matching and optical flow benchmarks, together with an extensive ablation study that analyzes the effectiveness of each component.

Strengths: The proposed layer GOCor can help alleviate the ambiguities in the matching field derived from the standard cross-correlation layer by increasing the uniqueness of the features extracted on the reference image and imposing regularities in the matching field induced by the query image. Repetitive patterns and textureless regions are properly addressed; also, the proposed objective is differentiable, which allows end-to-end learning. Good explanation and well-described implementation details. Moreover, promising results compared to the standard one.

Weaknesses: Even though the proposed method could disambiguate similar regions, the introduced internal optimization could also incur heavy computation. Should give some advice on how to reach a good speed-accuracy trade off.

Correctness: The main claims are supported by the experiments. But on the design of the regularization operator, it is called induced by the query frame? However the convolution kernel R_theta is only set of parameters trained by both the reference frame and the query frame. Does it actually take in the query frame as input? If not, why would call Eq 6 the query frame objective? Seem to me it is just a smoothing operator.

Clarity: Yes, with a nice suppmat.

Relation to Prior Work: Yes.

Reproducibility: Yes

Additional Feedback: I would like to keep my rating. Appropriate naming of the terms are still not well addressed in my opinion. But I think it does not hurt the contribution of the paper. If addressed, would be more elegant.

[Author Response · NeurIPS 2020]

We thank the reviewers for their positive feedback: well written paper (**R1**,**R2**,**R3**,**R4**) with clear structure and pictures (**R2**,**R3**), interesting method (**R1**,**R3**), useful contribution (**R2**), novelty of the method (**R3**) with well designed losses (**R2**,**R3**), extensive and thoughtful experiments (**R1**,**R2**), superior performance (**R3**) and promising results (**R1**,**R4**).

**[R1-Q1] Intuition about objectives:** In Sec. 3.5, we aim to provide intuition about our query frame objective Eq. (6) by relating it to classical optical flow methods in an extensive discussion (see L208-221). Regarding Sec. 3.4, we are not aware of similar attempts and thus included an extended description (L150-175) along with visualizations (Fig. 1,3).

**[R1-Q2a] Robustness to appearance and geometric variations:** As the reviewer points out, the value of additional reference frame information (L103-112) is not as pronounced in semantic matching, since images depict different scenes and object instances. As suggested by the reviewer, we nevertheless evaluate GOCor (without any retraining) for dense se-

Table 1: PCK [%] on TSS.

|  | FGD3Car | JODS | PASCAL | All |
|---|---|---|---|---|
| GLU-Net [49] | 93.2 | 73.3 | 71.1 | 79.2 |
| Semantic-GLU-Net [49] | 94.4 | 75.5 | 78.3 | 82.8 |
| GLU-Net-GOCor | **95.0** | **78.9** | **81.3** | **85.1** |

mantic matching on the TSS [Taniai, 2016] dataset in Tab. 1. In fact, our GLU-Net-GOCor sets a new state-of-the-art on this dataset, even outperforming Semantic-GLU-Net [49]. Moreover, the results for increasing view-point changes on ETH3D [43] (Fig. 4), indicate that GOCor better copes with large appearance and geometric variations.

**[R1-Q2b] Results without coarse-to-fine:** We perform a preliminary experiment by computing the flow directly from the global correlation through an argmax operation. Compared to feature correlation, our GOCor achieves 8.0% and 13.0% better EPE on HPatches [3] and KITTI-2015 [13] respectively. Importantly, the correspondence volume generated by GOCor is also much more discriminative (Fig. 1), greatly enhancing the results of correspondence networks.

**[R1-Q3] Using cross entropy in Eq. (5):** While our objective function and optimization module could be formulated with cross-entropy instead, our squared loss allows the use of Gauss-Newton for efficient optimization and flexibility through learned parametrization. We will consider this interesting suggestion for future work.

**[R1-Q4] Additional references:** We thank the reviewer for the references and will include them in the paper.

**[R2-Q1] Interchanging query and reference features:** Prior to submission, we experimented with also interchanging the two frames at the global level, and then fusing the two resulting GOCor correspondence volumes. However, we only observed marginal improvements. E.g., on KITTI-2015 it obtains an EPE of 11.07 and an F1 of 54.68% compared to 10.97 EPE and 55.62% F1 for the baseline 'BaseNet $L_r + L_q$' (suppl. Tab. 7). We will include this experiment.

**[R2-Q2] Nature of optimizer (L231-242):** It is an online process performed at every forward pass of the network.

**[R2-Q3, R3, R4-Q1] Computational complexity:** We perform a detailed analysis of the run-time for varying number of optimizer iterations in suppl. Tab. 2 (Sec. E.1) and Tab. 8, which we will move to the main paper. While our GOCor has an impact in run time, we believe that it is small compared to the improvement in performance brought by our module. In suppl. Sec. E.2 we discuss and suggest a good speed-accuracy trade-off.

**[R3-Q1] Properties of the regularizer (Sec. 3.5):** With our claim in L219 we mean that our formulation in Eq. (6) is capable of learning filters $R_\theta$ that can enforce local uniqueness (will be clarified). While we did not claim that it actually does, this is a very interesting question that is, however, difficult to verify empirically. In practice, we often observe that the query objective also has a 'peak-enhancing' effect, as shown in Fig. 1 below.

**[R3-Q2] Smoothness constraints on the flow field:** While this is an interesting point, it is very difficult to compare the two strategies in practice, since the flow in our approach is predicted with a deep CNN. As a result, a regularization loss on the flow itself cannot easily be embedded into our objective. We will nevertheless consider this for future study.

**[R3-Q3] Performance on occlusion data:** As shown in suppl. Tab. 5, in occluded regions ("EPE unmatched") of the Sintel test set, GOCor provides relative improvements of 6.25% and 2.16% on the clean and final pass respectively. On KITTI-2015, GOCor improves the performance of PWC-Net and GLU-Net in occluded regions, as shown in Tab. 2. Moreover, we did not observe noticeable blurring at occlusion boundaries.

Table 2: AEPE/F1 [%] on KITTI-2015.

|  | Not occluded | Occluded | All |
|---|---|---|---|
| GLU-Net | 4.67 / 27.83 | 21.95 / 67.44 | 7.49 / 33.83 |
| GLU-Net-GOCor | **4.22 / 22.03** | **19.07 / 58.61** | **6.68 / 27.57** |
| PWC-Net | 5.40 / 25.16 | 34.39 / 78.58 | 10.81 / 32.75 |
| PWC-Net-GOCor | **5.02 / 23.53** | **34.06 / 77.84** | **10.33 / 30.53** |

**[R4-Q2] Clarification of the design and name of the query frame objective:** We call Sec. 3.4 and 3.5 the reference and query frame objectives since they are evaluated based on the correspondence volume predicted on the reference and query frame respectively. In Eq. (6), the convolutional kernel $R_\theta$ is applied to the correspondence volume $C(w, f^q)$ between our filter map $w$ and the query feature map $f^q$. Note that it is the filter map $w$ that is optimized using Eq. (5) and (6) at each forward pass, while the kernel $R_\theta$ (L204) is learnt, along with all other network parameters, by the SGD-based minimization of the same final network training loss used in the GLU-Net and PWC-Net baselines.

Figure 1: Visualization of the matching confidences computed between the indicated location (green) in the reference image and all locations of the query image. We compare with and without utilizing the query frame objective $L_q$.

[Meta-Review · NeurIPS 2020]

All reviewers consider the paper novel and the results compelling. The work is also clearly presented. There are still few (optional) improvements that I would encourage the authors to consider: more detailed analysis of what happens in presence of conclusions and an ablation study on the query-frame objective as one reviewer raised the concern on whether it actually helps in practice.